# Ultra-bright Raman dots for multiplexed optical imaging

Zhilun Zhao [1,3], Chen Chen[1,3], Shixuan Wei[1], Hanqing Xiong[1], Fanghao Hu [1], Yupeng Miao [1], Tianwei Jin[2] & Wei Min [1✉]

Imaging the spatial distribution of biomolecules is at the core of modern biology. The development of fluorescence techniques has enabled researchers to investigate subcellular structures with nanometer precision. However, multiplexed imaging, i.e. observing complex biological networks and interactions, is mainly limited by the fundamental 'spectral crowding' of fluorescent materials. Raman spectroscopy-based methods, on the other hand, have a much greater spectral resolution, but often lack the required sensitivity for practical imaging of biomarkers. Addressing the pressing need for new Raman probes, herein we present a series of Raman-active nanoparticles (Rdots) that exhibit the combined advantages of ultra-brightness and compact sizes (~20 nm). When coupled with the emerging stimulated Raman scattering (SRS) microscopy, these Rdots are brighter than previously reported Raman-active organic probes by two to three orders of magnitude. We further obtain evidence supporting for SRS imaging of Rdots at single particle level. The compact size and ultra-brightness of Rdots allows immunostaining of specific protein targets (including cytoskeleton and low-abundant surface proteins) in mammalian cells and tissue slices with high imaging contrast. These Rdots thus offer a promising tool for a large range of studies on complex biological networks.

[1] Department of Chemistry, Columbia University, New York, NY, USA. [2] Department of Applied Physics and Applied Mathematics, Columbia University, New York, NY, USA. [3] These authors contributed equally: Zhilun Zhao, Chen Chen. ✉email: wm2256@columbia.edu

Imaging techniques are one of the essential methods in biological sciences. In particular, immunofluorescence microscopy (IFM), where fluorescent materials are coupled to affinity binders such as antibodies, is widely used to visualize the distribution of various biomarkers. Over the last two decades, optical advancements such as super-resolution microscopy have made IFM possible to resolve cellular structures with nanometer resolution[1–3]. Meanwhile, chemical advancements have led to the development of new fluorescent materials with superb photophysical properties, especially luminescent nanoparticles[4]. These nanoparticles offer considerable advantages of brightness and photostability, and have been successfully demonstrated in biological studies.

However, multiplexed imaging of a large number of molecular targets remains the next grand challenge. To address this challenge, some multiplexed IFM methods have been developed[5–10]. These methods involve repetitive staining, imaging, and antibody/fluorophore removal, which need to be carried out over multiple cycles in order to image multiple biomarkers. Such procedures take a prolonged time and face inevitable complications of imaging registration and epitope degradation over multiple staining rounds. The problem is rooted from the fundamental "spectral crowding" of fluorescence spectroscopy (typically ~50 nm linewidth), so that less than a few channels (4–5 'colors') can be imaged simultaneously.

Distinct from fluorescence, Raman scattering-based techniques circumvent the spectral crowding problem and offer an elegant strategy towards super-multiplexed optical imaging. Thanks to the much narrower (~50 times) linewidth of vibrational modes than fluorescence spectra, it is possible to have over 20 simultaneous channels with the Raman approach with the potential for further expansion. However, the Raman scattering cross-section is generally $10^{10}$–$10^{14}$ times smaller than that of fluorescence dyes, making it almost impractical for immunostaining with simple Raman probes. For this reason, various techniques have been developed to enhance the Raman signal. Stimulated Raman scattering microscopy (SRS)[11], for example, utilizes stimulated scattering effect to accelerate vibrational activation rates by up to $10^8$ times[12]. But the amplification with SRS alone is still not enough for immunostaining of specific proteins using common Raman probes.

Other approaches of amplifying Raman signal include surface enhanced Raman scattering (SERS) and Raman active polymers/nanoparticles. SERS depends on the localized surface plasmons to enhance the Raman signal by as much as $10^{11}$, and has been used widely for biosensing[13]. However, for immuno-SERS, it generally takes a prolonged time to acquire SERS spectra for each pixel, and the image resolution is low[14]. More importantly, SERS particles usually require relatively large particle sizes (>50 nm) for enough signal enhancement, which makes it challenging to overcome the diffusion barrier of biological samples and thus limiting the accessibility of SERS particles to the biomarkers of interest. Furthermore, SERS signal largely depends on the generation of hotspots, which is difficult to precisely control. Hence, despite the recent advance, immune-SERS has seldom achieved quantitative staining in practice[15–17].

Raman active polymer/nanoparticles incorporate a large number of vibrational probes (such as alkyne, nitrile, and C-D) covalently into the polymer backbone to increase Raman signal[18–20]. However, chemical synthesis or surface modification is complicated in the reported demonstrations, and as a result, it is challenging to create a large panel of materials with resolvable Raman spectra for multiplexed imaging purposes. It is also noticeable that, similar to SERS particles, these reported polymer nanoparticles are usually large (50–100 nm), which makes diffusing into the cell a great challenge. As a result, practical immunostaining has not been demonstrated inside cells[18–20].

Hence new Raman-based probes that take the advantage of narrow spectral features for multiplexed imaging are in great need. Two critical aspects shall be taken into account. The first consideration is the brightness of the labeling material, which is a recurring theme in Raman-based microscopy techniques. The second consideration is the size of the labeling material, as it is generally difficult for large probes to transport into crowded macromolecular environments in biological samples. The interior of cells appeared as porous media with a throat size of ~20 nm[21]. Imaging with quantum dots of various sizes also showed dramatic signal decrease as the hydrodynamic diameter increased[22,23]. Importantly, these two factors are inter-related. It is generally difficult for existing methods such as SERS and Raman active polymer/nanoparticles to fulfill both requirements at the same time: when these probes were made small enough, they often lack sufficient amplification factors for the required brightness[24,25]. Identifying this problem as the bottleneck of the multiplexed Raman imaging technology, we wish to develop a new strategy to meet both requirements.

Here, we present a new set of Raman-active nanoparticles (Rdots) that are both ultra-bright and compact to address the pressing demand for Raman probes. We took advantage of the large Raman cross-sections of the newly developed Carbow dyes[26] and other molecules containing the alkyne or nitrile group, and non-covalently incorporated them into compact polymeric nanoparticles (20 nm) through a simple swelling-diffusion approach (Fig. 1). These Rdots have several advantages over the existing methods. Due to the tight stacking of Raman probes inside nanoparticles, Rdots are of great brightness with relatively small size. When coupled SRS microscopy with Rdots, we achieved a detection limit down to pM scale in solution and obtained evidence supporting single particle imaging. This is more sensitive than all the other organic-based Raman reporters so far. We further validated cytoskeleton immunostaining in mammalian cells using correlated fluorescence and SRS microscopy. Thanks to their ultra-brightness and compactness, Rdots offered superb performance on the gold standard cytoskeleton and low-abundant surface proteins. Harnessing the narrow Raman spectra, we then demonstrated the potential of Rdots for multiplexed imaging in mammalian cells and tissue samples. These promising results suggest that Rdots can be useful for a variety of multiplex optical applications.

## Results

**A general method to prepare compact multi-color Raman polymeric nanoparticles.** Our goal is to develop Raman-based nanoparticles that are both bright and compact enough to facilitate a multiplexed imaging platform. To avoid the complication of synthesizing Raman active polymers covalently as in previous literature[18–20], we adopted and optimized a swelling-diffusion approach to incorporate many Raman probes into small nanoparticles[27–29]. Briefly, organic solvents such as tetrahydrofuran (THF) were first mixed with the solution of Raman probes, and then added to polymeric nanoparticles suspended in water. Since most Raman probes were hydrophobic and contained aromatic structures, we chose commercially available 20 nm polystyrene (PS) nanoparticles for their hydrophobicity and aromatic rings to have optimal interaction between Raman probes and nanoparticles matrix. After the addition of THF, PS nanoparticles were swelled so that Raman probes could diffuse into the PS nanoparticle matrix. Then the excessive amount of aqueous solution was added to shrink the nanoparticles back to the original size and trap the Raman probes inside (Fig. 1).

We chose six recently developed Carbow probes[26] to demonstrate for their large Raman cross sections (Fig. 2a). All

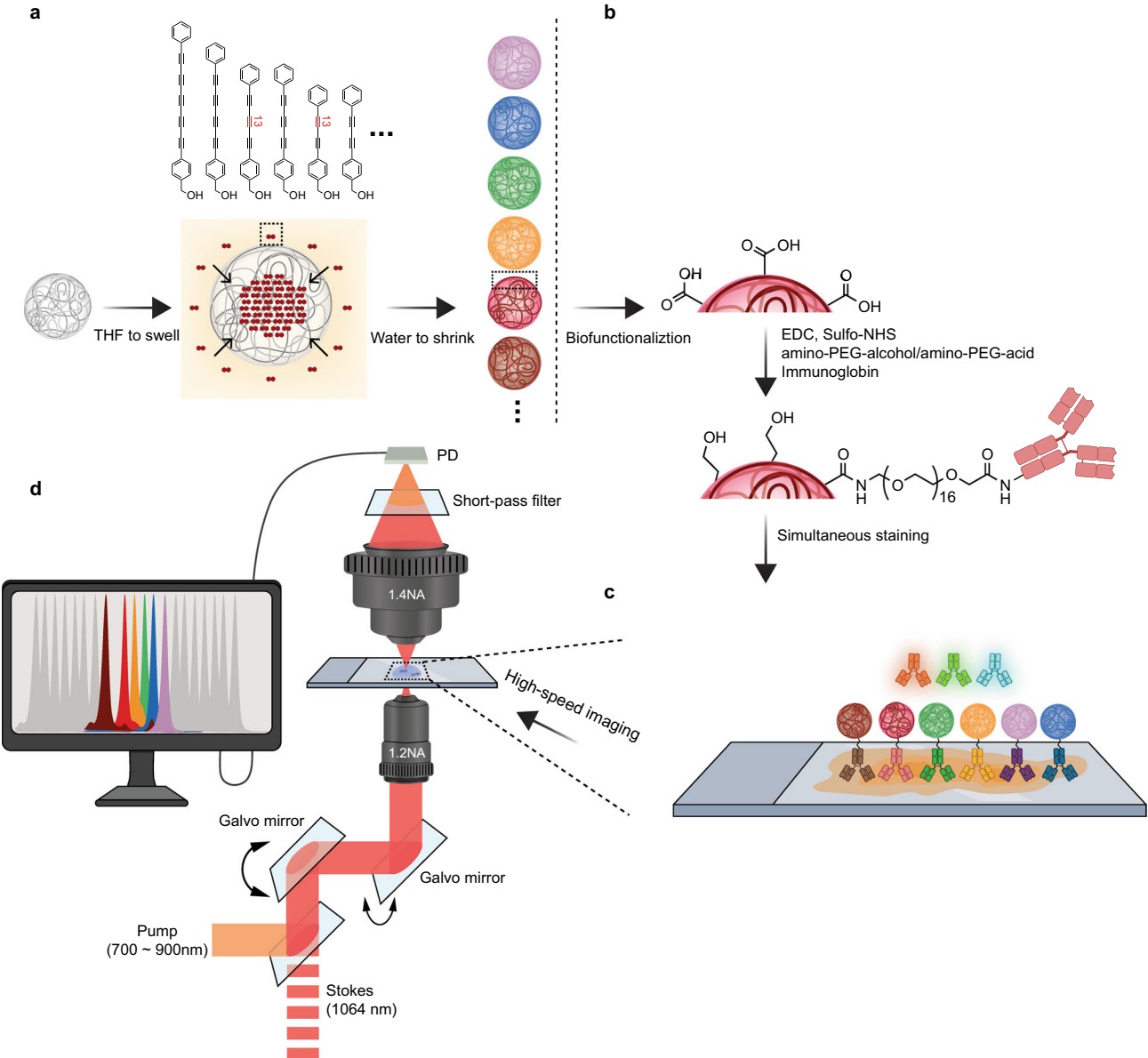

**Fig. 1 Schematics of preparing Rdots and their biofunctionalization. a** 20 nm polystyrene (PS) nanoparticles are first swelled in organic solvents such as tetrahydrofuran (THF), and small-molecule Raman probes are allowed to diffuse into the nanoparticles, followed by shrinking the nanoparticles to entrap the Raman probes to generate Rdots. The dense packing of Raman probes inside nanoparticles makes Rdots ultra bright. **b** For easy biofunctionalization, Rdots are first conjugated with a mixture of amine-PEG8-alcohol and amine-PEG16-acid through EDC/NHS coupling to react with abundant carboxyl groups on the surface. Such combination of the two PEG chains can help reduce the non-specific binding and aggregation. Then IgG or other amine bearing bio-molecules are conjugated to the carboxyl groups from PEG-acids. This two-step procedure helps to increase the hydrophilicity and to greatly reduce non-specific binding. **c** Simultaneous immunostaining of Rdots and fluorescence probes (shown in shaded antibodies). **d** Multiplexed imaging with stimulated Raman scattering (SRS) microscopy, thanks to the narrow peak width of Rdots.

the six Carbow probes were successfully incorporated into PS nanoparticles to generate a series of Rdots. As shown in Fig. 2a, these Rdots were spectrally resolvable, and had similar Raman intensity per nanoparticle (Supplementary Fig. 1 and Supplementary Table 1). We then acquired the Raman spectra of these Rdots and compared them with free Carbow probes dissolved in DMSO (Fig. 2b and Supplementary Fig. 2). Negligible spectral shift or broadening of Carbow probes was observed when they were incorporated inside Rdots. This makes it much easier to expand the 'color palette' because one only needs to develop and synthesize new small-molecule Raman probes and can maintain their spectral characteristics when using them to generate new kinds of Rdots. In addition, these Rdots were stable in aqueous

solution – we did not observe signal loss after 5 months since they were prepared, indicating no leakage of Raman dyes from the nanoparticles (Supplementary Fig. 1). This is consistent with the fact that both the PS nanoparticle matrix and Raman probes are hydrophobic, and it is thermodynamically favorable for them to minimize contact with the aqueous solutions. The doping procedure did not change the size of the original PS nanoparticles (Fig. 2e) measured by dynamic light scattering (DLS). The morphology was also confirmed unchanged with SEM (Supplementary Fig. 3).

Our method can be generalized to other Raman probes. We tested 19 common probes containing alkyne or nitrile moieties[30], and all of them were successfully incorporated into PS

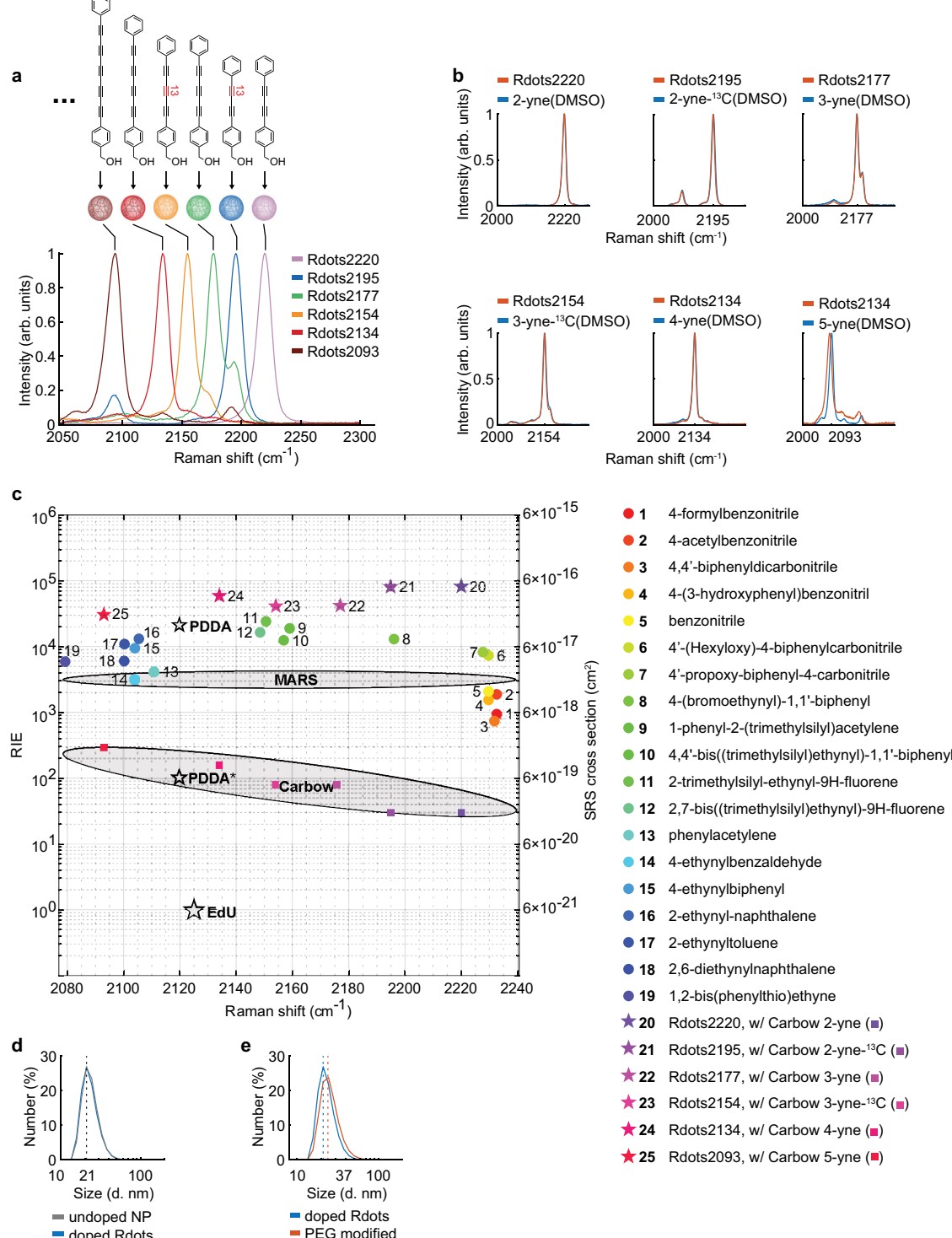

**Fig. 2 Characterizations of Rdots. a** Spontaneous Raman spectra of six Rdots generated using Carbow dyes. arb. units: arbitrary units. **b** Raman spectra of Rdots and free Carbow dyes in DMSO, showing no significant peak shift or broadening. arb. units: arbitrary units. **c** RIE and Raman shift of Rdots made by various alkyne and nitrile containing probes. Rdots were made with a large variety of compounds that contain alkyne or nitrile molecules, and their RIE (relative Raman intensity versus EdU) values and Raman shifts were measured. RIE values were measured with Spontaneous Raman spectrometer with 532 nm excitation wavelength, except MARS dye, which were evaluated with SRS microscopy. Probes indicated with colored stars were used for immunostaining in this study. PDDA: poly(deca-4,6-diynedioic acid)[20], EdU: 5-ethynyl-2'-deoxyuridine. **d** Hydrodynamic diameter of PS nanoparticles before and after doping measured by DLS. The size distributions show no change when PS nanoparticles are doped with Raman dyes, and the resulting 21 nm Rdots are monodisperse. **e** Hydrodynamic diameter of Rdots after biofunctionalization.

nanoparticles (Fig. 2c, molecular structures in Supplementary Table 2). No spectral broadening or significant shift was observed either. In total, 10 different Rdots with resolvable Raman peaks were generated with the selected Carbow dyes and Raman probes (Supplementary Fig. 4). In addition, although we demonstrated here with six selected Carbow dyes, the full panel of 20 Carbow dyes could all be used, making it possible to generate more than 20 kinds of fully resolvable Rdots.

**Dense volumetric packing of Raman probes enables ultra-high Raman brightness of Rdots.** The relative Raman intensity versus EdU values (RIE), which are commonly used for quantifying the 'brightness' of Raman probes, were acquired for all Rdots (Fig. 2c). Remarkably, most of the Rdots had RIE over $10^4$, making them the brightest organic-based Raman probes to date. It was also nearly four times brighter than the recently reported ultra-strong Raman active polymer PDDA[20] (Fig. 2c), even though PDDA takes advantage of the resonance Raman effect in the visible range (which was not compatible with the standard SRS microscopy operating in the near infrared). We then estimated the SRS cross section of each Rdots (Fig. 2c, see Supplementary Information for detail) by quantitatively comparing its molar intensity with that of methanol whose SRS cross section was reported. The SRS cross section of Rdots2220, for instance, was estimated to be as high as $5 \times 10^{-16}$ cm$^2$, which is more than 40 times larger than currently the best organic Raman dyes (MARS)[31], and a value even higher than the absorption cross section of a fluorescence dye.

We then seek the structural insight for the great RIE values and SRS cross sections of Rdots. Two independent methods were carried out to estimate the number of Raman probes per nanoparticle (see Supplementary Note for detail), and both of them yielded similar results. The number varies for different Raman probes (Supplementary Table 1), but in general the doping efficiency was surprisingly high. For example, $2.7 \times 10^3$ Carbow 2-yne were incorporated per 20 nm Rdots2220, which led to a local concentration of 1.2 M. At such a high local concentration, probes are densely packed and the average intermolecular distance is less than 1 nm. Note that fluorescence will be self-quenched at such a short distance, whereas Raman does not suffer from any quenching effect.

Such a dense packing is likely a result of strong intermolecular interactions between Raman probes and the polymer matrix. This makes sense chemically because both the polymer and the Raman probes are hydrophobic and contain aromatic structures, enabling hydrophobic interaction and π-π stacking. Hence, the resulting dense volumetric packing of Raman probes inside Rdots manifests the greatest brightness among all organic-based Raman reporters so far.

**Evidence supporting single particle detection and imaging of Rdots with SRS microscopy.** Encouraged by the ultra-brightness of Rdots, we next sought to characterize the detection limit using the emerging SRS microscopy to take advantage of the high speed and spatial resolution of SRS. Superb detection sensitivity and spectral specificity were achieved with a pico-second pulse laser in our SRS microscope system. A linear relationship between the SRS signal and the concentration of Rdots was observed (Fig. 3a). The shot-noise limited detection limit was estimated[32] (Fig. 3a, red dash line), which was defined when the signal to noise ratio (SNR) was equal to 1. Based on the detection sensitivity of our imaging platform, the detection limit of Rdots was estimated to be $900 \pm 50$ pM ($\pm 99\%$ confidence interval). Indeed, this is nearly two to three orders of magnitude more sensitive than those from previously reported organic Raman probes[26,33].

At this sub-nanomolar detection limit level, single particle detection limit might also be possible, as the average number of particles within the laser focal volume ($\sim 2 \times 10^{-16}$ L) is less than 1. Indeed, after we embedded Rdots in agarose gel and took SRS images of the immobilized Rdots, we observed Raman signals from diffraction limited spots whose intensity was consistent with our theoretical estimation from a single particle (Fig. 3b). SRS spectra were also acquired from these 'single particles' by tuning the pump laser beam (Fig. 3c), consistent with the solution spectra. However, unlike single-molecule fluorescence whose stoichiometry can be determined by observing the abrupt photobleaching steps from individual fluorophores, it is difficult to confirm whether the signal from diffraction limited spots in the SRS image is indeed from single particles or from the aggregation of a few single particles with the conventional photobleaching method. Hence, we seek another way to validate single-particle detection sensitivity. Our new strategy is to record a large number of potentially single particle events and analyze the statistics of their SRS intensity distribution. If single particles can be indeed observed, we shall be expecting to observe a quantized intensity distribution, from single-particle, double-particle, triple-particle, and so on.

We then conducted such a systematic statistical validation. Since the colloidal stability was primarily maintained by the negatively charged carboxyl groups on the surface of Rdots, we first quantified the intensity distribution at a more basic condition (pH = 8.5) so that particle aggregates were less likely to form due to the electrostatic repulsion from the deprotonated carboxyl groups. We observed two evident peaks in the intensity distribution histogram (Fig. 3d), and the majority of the potential single particle events reside in the less intense peak. The particles corresponding to the second peak have double the SRS intensity as the first peak has, suggesting these are double-particle aggregates. No additional distribution peak beyond was observed at this basic pH condition, indicating that most Rdots dispersed as single particles, a small portion as double particles, and none as triple particles. We then acquired the intensity distribution at a more acidic condition (pH = 6.0). Some of the surface carboxyl groups were protonated at this pH, resulting in less electrostatic repulsion and Rdots should be more likely to form aggregates. Consistent to our hypothesis, more double particles were formed and some Rdots even formed triple-particle aggregates (Fig. 3e). Similar to Fig. 3d, the intensity distribution is well quantized, with the triple particles being three times more intense than the single particles.

Additionally, Fig. 3f shows an SRS image where single-particle aggregates (spot 1), double-particle aggregates (spot 2), triple-particle aggregates (spot 3), and quadruple-particle aggregates (spot 4) appear in the same field of view. Line profiles shown in Fig. 3g across these particles and particle aggregates quantitatively indicate that spots 2–4 had signal intensities that are integer multiples of the single-particle spot 1. If the minimal aggregates were to start from 2 particles for example, then the second 'dimmest' spots consisting of 3 particles would be 3/2 of the brightness. Yet, we did not observe this in either the histogram (Fig. 3d) or the single-particle line profiles (Fig. 3g). This suggests that the 'dimmest' spots are indeed single particles that exhibit elementary signal intensity.

The analysis of the signal to noise ratio also suggests single-particle detectability. Our previous study showed Carbow 2yne has a detection limit around 4.3 μM[26], which is equivalent to 250–500 molecules within the excitation volume. As mentioned earlier, there were estimated $2.7 \times 10^3$ Carbow 2-yne molecules entrapped in each Rdots2220 nanoparticle measured in solution (in which the particles were monodisperse), so the expected SNR from single Rdots2220 is around 8. Indeed, under SRS imaging,

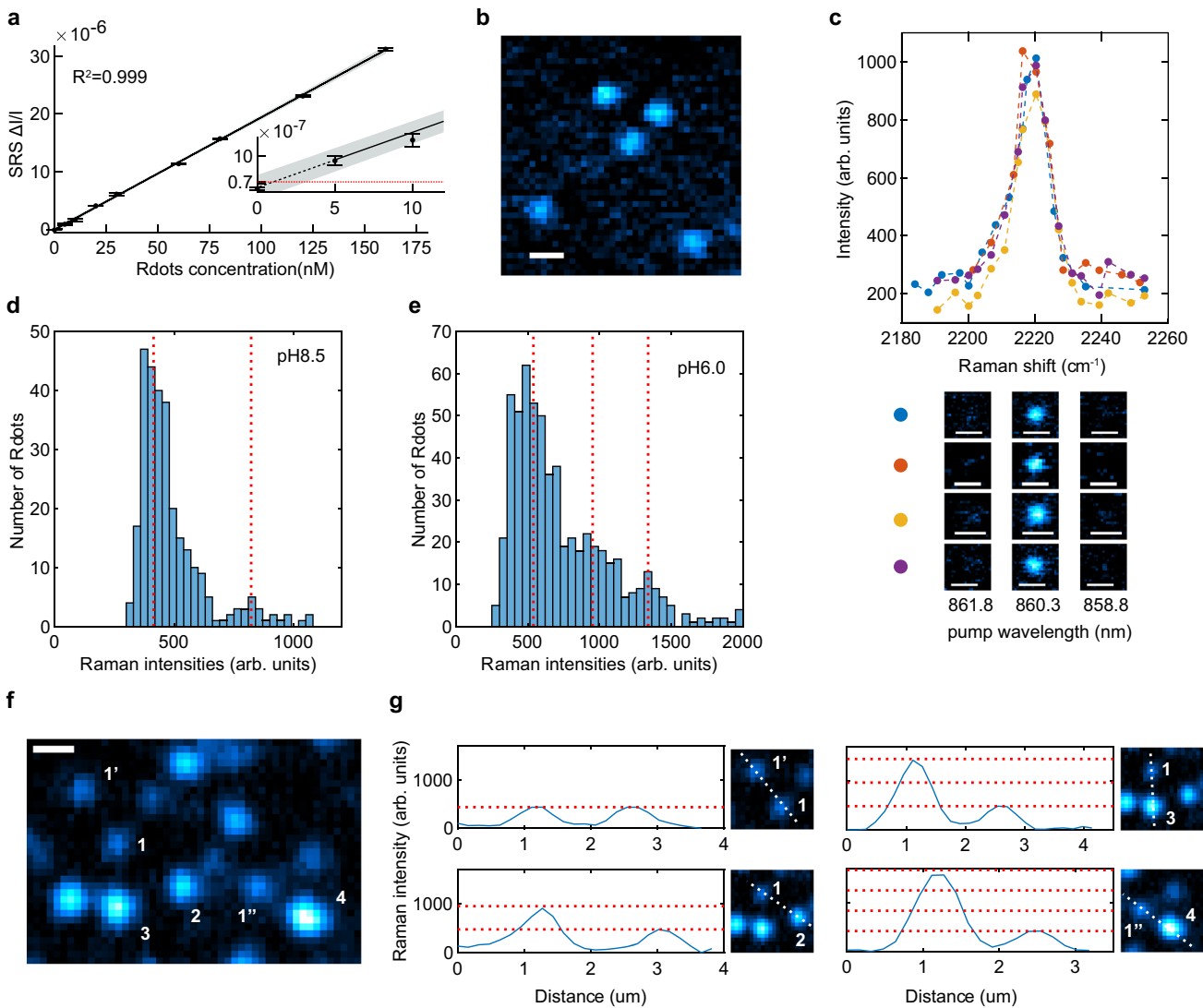

**Fig. 3 Evidence for single-particle sensitivity of Rdots under SRS microscopy. a** Linear concentration dependence of Rdots with sub-nM SRS detection sensitivity. Red dashed line in the insert indicates shot-noise-limited SRS detection limit where SNR = 1. Error bars, mean ± s.d.; $n = 3$ measurements. Solid line shows a linear fitting ($R^2 = 0.999$), shaded area indicates 95% confidence interval of the linear fitting. arb. units: arbitrary units. **b** A representative image of potential single-particle spots. Scale bar: 1 μm. **c** Raman spectra of 'single' Rdots2220 acquired by SRS microscopy by sweeping the pump wavelength. The original SRS images below at three marked pump wavelengths show the signal can be tuned off by only 1.5 nm. The spectra are in consistence with bulk measurements of Rdots by spontaneous Raman spectroscopy. Scale bar, 1 μm. arb. units: arbitrary units. **d** SRS intensity distribution of individual spots at pH 8.5. Red dash lines indicate fitted peak positions. The quantized distribution suggests spots corresponding to single-particle and double-particle aggregates. $n = 289$ from 7 replicates. **e** SRS intensity distribution of individual spots at pH 6.0. Red dash lines indicate fitted peak positions. The quantized distribution suggests spots corresponding to single-particles, double-particle aggregates, and triple-particle aggregates. $n = 595$ from 6 replicates. arb. units: arbitrary units. **f** A representative image of spots from single-particles and multi-particle aggregates. 1, 1', and 1": single-particle; 2: double-particle; 3: triple-particle; 4: quadruple-particle. Scale bar: 1 μm. **g** Line profiles across spots in **f**. Quantized signal intensity is evidently observed. arb. units: arbitrary units.

the SNR of the 'dimmest' single-particle candidates is ~8. On the other hand, the high SNR of the 'dimmest' imaged spots shown in Fig. 3 is much greater than the detection limit. This means if there were a spot of lower intensity, we would have been able to detect it. Yet we did not find additional peaks of weaker signal in the histograms (Fig. 3d, e), nor in the single-particle images (Fig. 3b, f). Taken together, these results strongly suggest that SRS microscopy enables single-particle detection and imaging of Rdots.

**Rdots enable immunostaining in cells and tissues.** Lastly, we aimed to biofunctionalize the newly developed Rdots for

biological imaging. We made use of the abundant carboxyl groups on the Rdots surface and firstly conjugated amine-PEG8-alcohol and amine-PEG16-acid with EDC/NHS coupling chemistry. This combination of PEGs reduced the surface charge without affecting the colloidal stability, and helped to minimize non-specific binding[34–36]. We then conjugated the secondary antibodies or protein A to the carboxyl groups of the PEG chain with the same coupling chemistry. DLS analysis showed that the PEG coating had only minor effects on overall Rdots size (Fig. 2f, the diameter of Rdots after PEG modification was ~23 nm). The relatively small size was comparable to that of quantum dots (10–20 nm), and smaller than most of the commonly used SERS nanoparticles (>40 nm) as well as recently reported Raman active

polymers (50–100 nm), making Rdots suitable for staining subcellular structures[4].

We started by examining the non-specific binding of Rdots. Secondary antibody-conjugated Rdots2220 were incubated with cultured mammalian cells without the use of any primary antibody. Only a negligible level of non-specific binding was observed (Fig. 4a, b). Raman peak at 2940 cm$^{-1}$ from C–H vibrations in proteins and lipids was used to locate the cells (Fig. 4a). Cytoskeleton proteins such as tubulin and vimentin are regarded as the gold standard targets of cellular immunostaining. We then visualized α-tubulin distribution patterns in Cos-7 cells using the α-tubulin primary antibody and secondary antibody conjugated Rdots2220 (Fig. 4c, d). The SRS image clearly shows the expected fine structures of microtubule filament networks (Fig. 4c). Tuning the pump laser wavelength by only 3 nm showed a negligible background signal in the off-resonance SRS channel (Fig. 4d). Such high spectral resolution is not achievable by fluorescence approaches. For SRS imaging, we achieved satisfactory quality with 10–30 us time constant and the same pixel dwell time, and it took less than 8 s to acquire an image of 512 by 512 pixels.

We then validated the immunostaining pattern of Rdots with correlated fluorescence and SRS microscopy. Cos-7 cells were first incubated with primary antibodies, and then stained by Rdots and fluorescent secondary antibodies sequentially. The resulting correlative images of α-tubulin staining (Fig. 4e–g, k–m) and of vimentin staining (Fig. 4h–j, n–p) show identical staining pattern between fluorescence images (Fig. 4e, h, k, n for the magnified view) and SRS images from Rdots (Fig. 4f, i, l, o for the magnified view). Colocalization analysis indicates a good correlation between the fluorescence channel and SRS channel with Pearson's $r$ greater than 0.8 for both microtubule and vimentin staining (Fig. 4q, s). Line profiles were also plotted across individual α-tubulin filaments (Fig. 4r) and vimentin filament (Fig. 4t) for both fluorescence channel and SRS channel. The two channels show very similar profiles, indicating immunostaining with Rdots had the same results as with conventional immunofluorescence staining.

We then explored immunostaining with Rdots for tissue samples. Epithelial cadherin (e-cadherin), which is a type of cell membrane-associated glycoproteins that mediate specific cell–cell adhesion, is exclusively expressed in epithelial cells[37]. Because of its unique spatial expression pattern, we used it as the target to demonstrate the sensitivity and specificity of Rdots immunostaining in tissue samples. In this article we used mouse colon frozen sections for their accessibility and popularity in clinical immunohistochemistry. Similar to the cell cytoskeleton staining, we obtained images acquired with conventional immunofluorescence and Rdots, respectively (Fig. 4u–w). The immunofluorescence image (Fig. 4u) shows e-cadherin staining pattern only in epithelial cells that are close to the lumen, but not in the connective tissue (submucosa) or the muscle layer (muscularis externa) surrounding the tissue slice, consistent with its histology. The SRS image of a different slice stained with Rdots2220 shows a similar staining pattern (Fig. 4v). Neither submucosa nor muscularis externa is stained positive for e-cadherin. Additionally, a magnified view of the Rdots stained sample (Fig. 4w) shows that only the cell membrane of those epithelial cells was stained, which is consistent with the expected subcellular distribution of e-cadherin. These results indicate the superb specificity of Rdots staining in tissue samples.

**Superb imaging detection sensitivity of immunostaining with Rdots**. Thanks to their ultra-brightness and compact size, SRS imaging of Rdots offers superb imaging detection sensitivity of immunostaining, based on several lines of evidence. First, for gold standard cytoskeleton proteins, the images (Fig. 4) by secondary antibodies functionalized Rdots exhibit a higher signal to noise ratio than previous reports using MARS or Carbow probes. Other Raman nanoparticles have not been demonstrated in imaging these gold standard targets. Second, it is well known that the resulting signal intensity could be amplified for 3–5 times by using secondary antibodies. However, multiplicity will be limited if secondary antibodies must be used. For the reason, we also acquired immunostaining on microtubule with primary antibodies functionalized Rdots (Supplementary Fig. 5a). The result confirms that Rdots are sensitive enough for immunostaining with only primary antibodies, which is very difficult with MARS or Carbow probes. Third, to further demonstrate sensitive imaging towards low abundant protein targets, we also stained and imaged CD44, a membrane marker, using Rdots2220 directly bioconjugated to anti-CD44 primary antibody. We observed strong membrane distributed Raman signal (Supplementary Fig. 5b), suggesting that Rdots are sensitive enough to visualize even low abundant membrane proteins. In contrast, the signal from the brightest MARS probe is much weaker for the same target (Supplementary Fig. 5c).

**Demonstration of the potential of Rdots for multiplexed imaging**. Due to the narrow Raman spectral features, multiple Raman probes with different Raman frequencies can be easily separated, resulting in high multiplexing capabilities. Indeed, multiplexed imaging has been well demonstrated by various Raman-based methods, including SERS[38,39]. Two to four channel imaging has been simultaneously detected with SERS measurements[39–41]. Given that Rdots inherit the same narrow Raman spectral features as other Raman-based probes and that SRS imaging of Rdots is compatible with additional fluorescence channels, we sought to demonstrate the potential of multiplexed imaging with Rdots.

After validating the staining pattern individually, we then used Rdots to demonstrate multiplexed imaging of microtubule, intermediate filaments and microfilaments. Cos-7 cells were first incubated with mouse-anti-a-tubulin and rabbit-anti-vimentin antibodies, followed by incubation with goat-anti-mouse conjugated Rdots2177, goat-anti-rabbit conjugated Rdots2220, and alexa647 labeled phalloidin. We chose these Rdots for their close Raman frequencies to demonstrate the fine spectral resolvability of our technique.

As expected, the imaging results clearly show spatial distribution of the three main kinds of cytoskeletal filaments with distinctive characteristics (Fig. 5a–d, e–h for magnified view). Minimal cross-talk was observed in all channels. The similar level of SNR was found in this three-color image compared to results where the cytoskeletal filaments were stained individually, indicating no significant competition between Rdots.

For our narrowband SRS imaging platform, we must acquire images at one Raman frequency at a time and then scan the next one. Thus, it was necessary for labeling materials to be photostable enough during the extended imaging procedure. We tested the photobleaching characteristics of Rdots, and found they were stable for at least 20 frames of continuous scanning (Supplementary Fig. 6). Only less than 10% signal loss was found at the end of this extended duration.

Lastly, we acquired four-color imaging of microtubule stained by Rdots2177 (Fig. 5i), intermediate filaments stained by Rdots2220 (Fig. 5j), microfilaments stained by alexa647 labeled phalloidin (Fig. 5k), and nucleus stained by NucGreen (Fig. 5l). The Rdots channels did not interfere with the additional nuclear staining by fluorescent dyes, indicating good compatibility of Rdots with fluorescence dye-based fluorescence imaging.

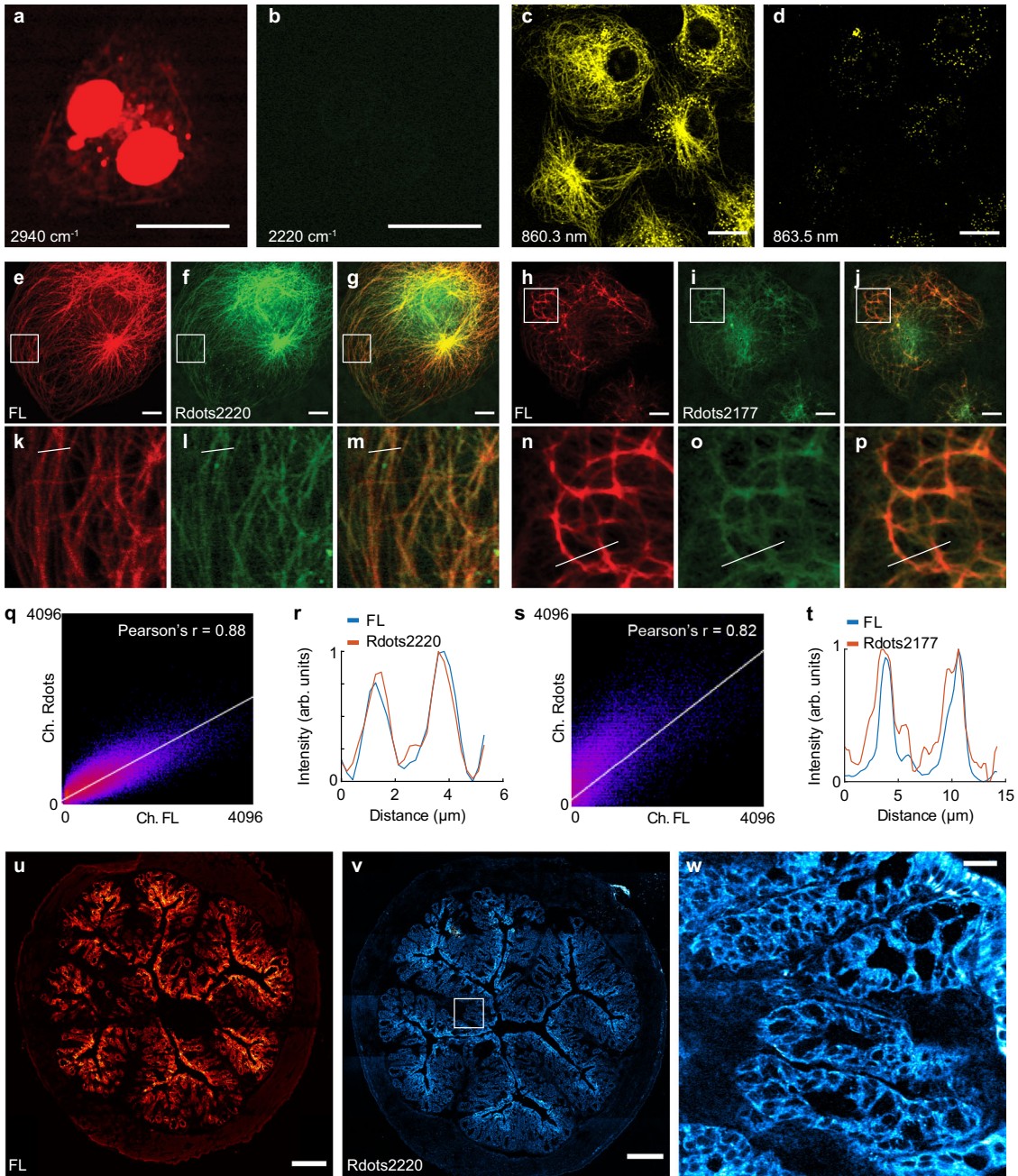

**Fig. 4 Immunostaining and imaging by Rdots with correlative fluorescence and SRS microscopy. a–b** Non-specific binding of Rdots. Imaging of C–H stretching mode (**a**) confirms the existence of the cell. No significant non-specific binding of Rdots2220 is observed (**b**). Scale bar: 20 μm. **c–d** Immunostaining and imaging of α-tubulin in fixed HeLa cells with Rdots2220 (**c**). The off-resonance image at 3 nm away shows negligible background (**d**). The brightness of **c** and **d** is set to be the same. Scale bar: 20 μm. **e–g** Correlative fluorescence and SRS immunostaining and imaging of α-tubulin in fixed Cos-7 cells with Rdots2220. **e** Fluorescence channel, **f** SRS channel, and **g** composite view of the two channels. Scale bar: 20 μm. **h–j** Correlative fluorescence and SRS immunostaining and imaging of vimentin in fixed Cos-7 cells with Rdots2177. **h** Fluorescence channel, **i** SRS channel, and **j** composite view of the two channels. Scale bar: 20 μm. **k–m** Magnified view of **e–g**. **n–p** Magnified view of **h–j**. **q** Colocalization analysis indicates good correlation between immunofluorescence and Rdots staining for α-tubulin (Pearson's $r = 0.88$). **r** Line profiles of the fluorescence channel (blue) and SRS channel (red) of microtubule filament shown in **m**. arb. units: arbitrary units. **s** Colocalization analysis indicates good correlation between immunofluorescence and Rdots staining for vimentin (Pearson's $r = 0.82$). **t** Line profiles of the fluorescence channel (blue) and SRS channel (red) of intermediate filament shown in **p**. arb. units: arbitrary units. **u** Immunofluorescence imaging of e-cadherin in the mouse frozen tissue section. **v** Immunostaining and imaging of e-cadherin with Rdots2220 in the mouse frozen tissue section. Scale bar: 400 μm. **w** Magnified view of **u**, the signals are only observed on the epithelial cell membrane. Scale bar: 50 μm.

## Discussion

Here we developed a general method to generate a series of compact Raman-active nanoparticles (Rdots) with ultrahigh brightness to meet the demand for smaller yet brighter Raman probes. The achieved detection sensitivity under SRS microscope is more sensitive than all the other organic-based Raman reporters so far. We have demonstrated successful immunostaining in both cell and tissue samples. The fine structures of cytoskeleton

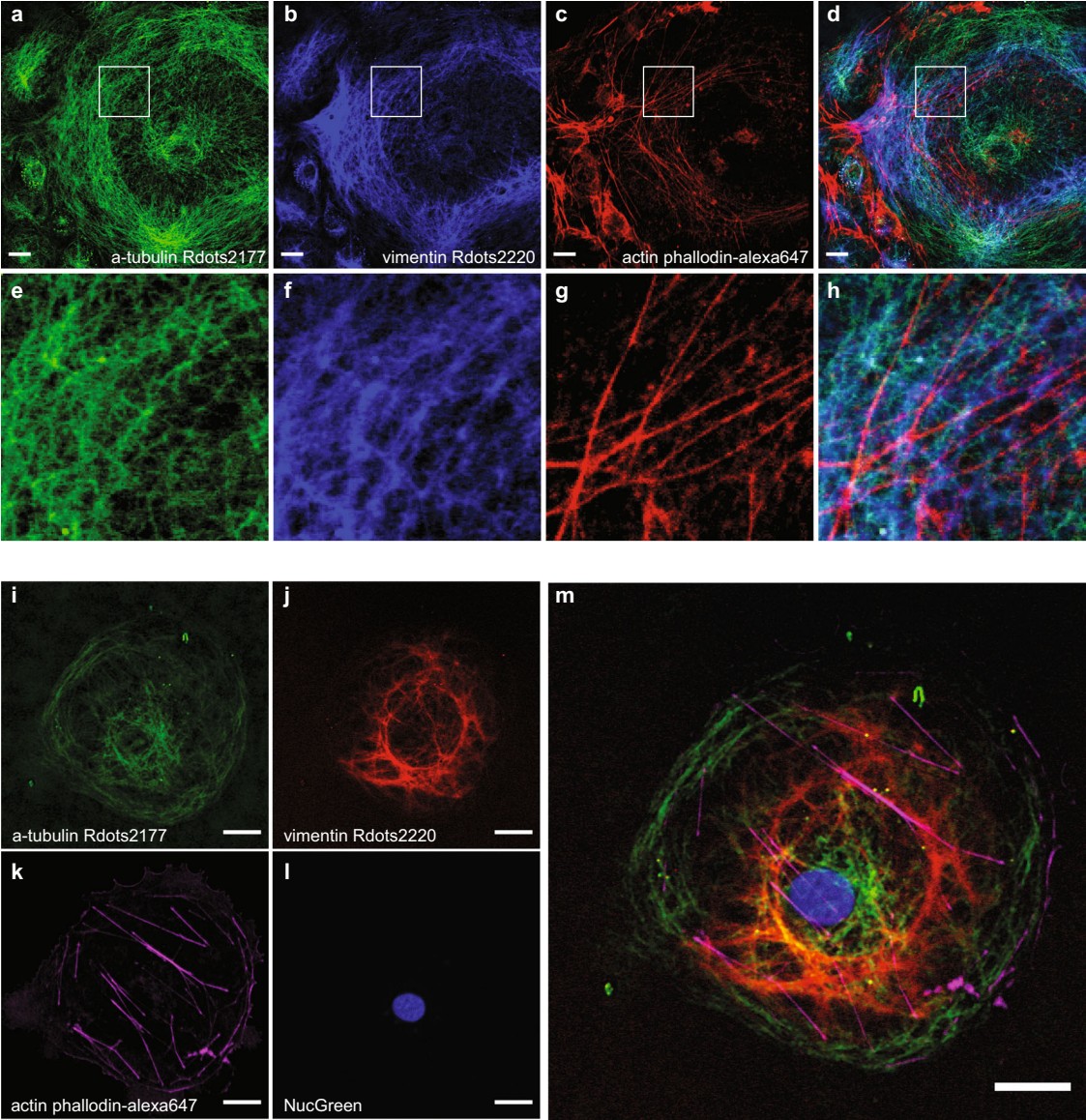

**Fig. 5 Multiplexed imaging with Rdots and fluorescence. a** α-tubulin immunostaining with Rdots2177. **b** Vimentin immunostaining with Rdots2220. **c** Actin staining with phalloidin-alexa647. **d** Composite view of the three channels. **e–h** Magnified view of **a–d**, respectively. **i** α-tubulin immunostaining with Rdots2177. **j** Vimentin immunostaining with Rdots2220. **k** Actin staining with phalloidin-alexa647. **l** DNA staining with NucGreen. **m** Composite view of the four channels. Scale bar: 20 μm.

filaments were visualized and verified with conventional IFM. Additionally, we also demonstrated immunostaining with primary antibody alone for microtubule visualization (Supplementary Fig. 5a) as well as low abundant membrane proteins (Supplementary Fig. 5b), which benefits from high detection sensitivity and compact size of Rdots. Simultaneous multicolor imaging was also demonstrated, supporting the potential of Rdots for multiplexed application.

The swell-diffusion approach we used to generate Rdots exhibit several advantages. First, one can incorporate a large number of small-molecule Raman probes into the nanoparticles. This leads to the ultra-high brightness of Rdots with the largest scattering cross sections of all reported Raman polymer nanoparticles, as well as to relatively small size of Rdots. Second, unlike SERS, the brightness of Rdots is only related to the number of incorporated Raman probes, but not to the surface morphology, and therefore a homogenous distribution of Raman signal per particles can be easily achieved (Fig. S4). This greatly facilitates quantitative

imaging using Rdots. Third, the swell-diffusion approach can be used for a wide selection of small Raman probes to generate corresponding Rdots, which can accelerate the development of new probes with unique Raman frequencies. It is also worth noting that, although some small molecules have smaller Raman cross sections, the brightness of the resulting Rdots is less affected (Fig. 2c and Supplementary Table 1) likely due to the greater number of small molecules being incorporated into the nanoparticle. Last but not the least, our method to make Rdots is easy-to-follow and can be easily scaled up. It does not require covalently labeling nanoparticles with Raman active small molecules, thereby avoiding the complications of engineering and synthesizing polymers. Although small-molecule Raman probes are trapped in Rdots through non-covalent interaction, the resulting Rdots can maintain their full brightness for at least 5 months.

A major challenge for the use of nanoparticles for imaging is their relatively large size compared to small organic dyes. Previous studies have found the diffusion barrier due to the large size

of nanoparticles and condensed environment in biological samples[22,23]. But we did not observe significant difficulty to achieve satisfactory imaging results, especially in tissue samples. This could be firstly due to the fact that we were using relatively small nanoparticles (~20 nm), which minimizes the diffusion barrier. Secondly for tissue samples, we used thin sectioned tissues that were ~8 μm thick. At this thickness, most of the cells in the sample were likely to be cut, exposing their subcellular targets for Rdots to bind[42]. Nevertheless, it is always desirable to make even smaller nanoparticles, especially when doing so could potentially increase specific surface area to improve the affinity of nanoparticles.

Artifacts have been frequently observed with other nanoparticle-based methods. The most common ones are nanoparticles aggregates and non-specific binding[4]. However, we did not observe significant aggregates or non-specific binding with Rdots. It might be because we used nanoparticles with abundant surface carboxyl groups to begin with, and the associated negative charges provide strong repulsive forces to prevent aggregates during the doping procedure. We then coated the Rdots with PEG, which reduces the negative charges and increases the hydrophilicity. Given that most non-specific binding happens due to the electrostatic interaction and hydrophobic interaction[35], the PEG coating has been proved to reduce both interactions and thus alleviate non-specific binding. The increased surface hydrophilicity, in turn, reduces the likelihood for Rdots to form aggregates, which is mainly a result from hydrophobic interaction in aqueous solution. Another artifact might be the sample burning due to local heating since a large number of vibrational modes inside Rdots are actively excited. However, we have not observed sample burning of Rdots during our imaging experiment. There may be two major reasons. First, Rdots are small nanoparticles, whose surface to volume ratio is high so that the heat generated can be efficiently dissipated. Second, water has large specific heat capacity and conductivity that dissipates the heat quickly. Since all the proposed applications require samples to be submerged in aqueous solutions, sample burning is less likely a concern.

While we demonstrated immunostaining with Rdots in this work, their applications can be much more general. Owing to the straightforward surface chemistry for biofunctionalization, a variety of molecules can be conjugated to Rdots. For example, amine-modified DNA strands can be attached to perform in situ hybridization, and the large multiplex capacity could greatly facilitate applications such as in situ transcriptome studies. Rdots can also be coupled with other high-through Raman microscopy methods such as flow cytometry to achieve highly multiplexed profiling of specific surface markers in cells[43]. Additionally, Rdots can be used with expansion microscopy as well to enable highly multiplexed super-resolution imaging[44], since one can still achieve good enough SNR even after signal 'dilution' due to the expansion. Rdots can also be used with hyperspectral SRS imaging platform to further increase the acquisition speed[45]. In this sense, we anticipate that Rdots will prevail and become an impactful and powerful tool.

## Methods

**Stimulated Raman scattering microscopy.** The stimulated Raman scattering microscopy was set up as previously described. In brief, an integrated picosecond laser system (Applied Physics & Electronics, Inc., picoEMERALD) was coupled into an inverted laser-scanning microscope (Olympus, FV1200). The Stokes beam (1064 nm, 6 ps pulse width) was intensity modulated at 8 MHz by an electro-optic-modulator, and a tunable pump beam (720–990 nm, 5–6 ps pulse width) was produced by a built-in optical parametric oscillator. The laser beams were focused on the sample through a 25x water immersion objective (Olympus, XLPlan N, 1.05 NA MP). The stimulated Raman loss signal was extracted from the pump beam by

demodulation at the 8 MHz frequency with near-short-noise-limited sensitivity using a lock-in amplifier (Zurich Instruments, HF2).

For the titration curve measurement, 150 mW pump and Stokes power were used, with 1 ms time constant. For immunostaining for cells and tissue, 100 mW pump and 150 mW Stokes power were used, with 10–30 us time constant and the matching pixel dwell time.

**Preparation of Rdots.** In a 5 ml tube, 320 μl 4% w/v polystyrene nanoparticles (Invitrogen, C37231) and 320 μl RO water were mixed. In another 1.5 ml tube, 40 μl 20x Carbow dye DMSO stock solution and 120 μl THF (Sigma, 401757) were mixed (final concentrations can be found in Supplementary Table 4). Then the THF/dye mixture was added drop by drop to the diluted nanoparticles to swell the nanoparticles and incorporate the dyes, followed by brief vortexing and gentle agitation with a rotary wheel for 30 min at room temperature. Next, 3 ml 20 mM phosphate buffer (pH 7.3) was slowly added to shrink and trap the dyes in the nanoparticles. The mixture was then centrifuged at max speed for 2 min to remove any insoluble materials, and the supernatant was collected to yield monodispersive Rdots. The Rdots were washed 3 times using Amicon 30 kDa MWCO filters (Millipore, UFC9030) with at least 10 ml RO water each to remove excessive organic solvents, and reduced the volume of nanoparticle colloidal suspension to ~300 μl. An optional centrifuge at 16,000 x g was performed to remove any precipitates. The exact concentration was determined by quantitatively measuring the 3054 cm$^{-1}$ Raman peak against the stock polystyrene nanoparticle solution with known concentration.

For the characterization of Rdots, spontaneous Raman spectra were acquired with a Horiba XploRA microspectrometer or a home-built Raman microspectrometer. The spectrometers were carefully calibrated with 50/50 (v/v) toluene/acetonitrile according to ASTM E1840-96. For DLS measurement, ~0.01% (w/v) nanoparticles were suspended in 10 mM pH 7.0 phosphate buffer.

**Single-particle imaging of Rdots.** Rdots were first diluted in 25 mM MES buffer pH 6.0 or 25 mM borate buffer pH 8.2 to the final concentration around 2 nM. 6% agarose gel was made by dissolving the right amount of agarose powder (A4018, Sigma) in the same buffer used for diluting Rdots. The mixture was allowed to gel at 95 °C on a heat block. On a pre-heated coverslip, 10 μl agarose gel and 1 μl diluted Rdots were quickly added, and were briefly mixed by pipetting a few times. Then a slide was immediately pressed on top to flatten the gel and a sandwiched sample was made ready for imaging.

**Biofunctioning of Rdots.** To functionalize Rdots with PEG polymer, first, 50 μl 4% Rdots were diluted to 450 μl 25 mM MES pH 6.0 buffer. Then to the diluted Rdots, freshly made 5 μl 1 M EDC (Thermo Scientific, 22980) and 25 μl 1 M sulfo-NHS (Sigma, 56485) in MES buffer was slowly added, quickly mixed by vortexing and kept gentle agitation with a rotary wheel for 30 min at room temperature. Next, excessive EDC/NHS was removed by washing two times with Amicon 30 kDa MWCO filters (Millipore, UFC5030) in 10 mM MES pH6.0 buffer, and the total volume was reduced to ~50 μl. 25 mM amino-PEG8-OH (Broadpharm, BP-21502) and 2.5 mM amino-PEG16-COOH (Broadpharm, BP-21880) solution mixture was freshly made in 500 μl 50 mM pH8.5 borate buffer, and mixed with the EDC/NHS activated Rdots and PEG mixture, followed by gentle agitation with a rotary wheel for 3 h at room temperature. After that, ethanolamine (Sigma, E0135) was added to a final concentration of 30 mM and incubated for 30 min at room temperature to quench the reaction. Finally, the functionalized Rdots were washed at least 5 times with Amicon 30 kDa MWCO filters (Millipore, UFC5030) to remove excessive reactants with 10 mM pH8.5 borate buffer or PBS buffer. If a longer PEG chain was used, more washes would be needed in order to completely remove the excessive.

To bioconjugate Rdots with IgG, proteinA (SpA) or other amine-bearing biomolecules, first, 50 μl 2% PEG-functionalized Rdots were diluted to 450 μl 25 mM MES pH6.0 buffer. Then to the diluted Rdots, freshly made 5 μl 1 M EDC and 25 μl 1 M sulfo-NHS in MES buffer were slowly added, quickly mixed by vortexing and kept gentle agitation with a rotary wheel for 30 min at room temperature. Next, excessive EDC/NHS were removed by washing two times with Amicon 30 kDa MWCO filters (Millipore, UFC5030) in 10 mM MES pH6.0 buffer, and the total volume was reduced to ~50 μl. IgG was dilute in 50 mM pH8.5 borate buffer so that the final concentration is 1–2 mg/ml. The EDC/NHS activated Rdots were then mixed with IgG, and kept gentle agitation with a rotary wheel for 3 h at room temperature. After that, ethanolamine was added to a final concentration of 30 mM and incubated for 30 min at room temperature to quench the reaction. The bioconjugated Rdots were purified with Sephacryl 300-HR (GE, S300HR) size exclusion chromatography column and PBST (0.05% tween-20) as eluant. The total volume was lastly reduced to the desired concentration with Amicon 30 kDa MWCO filters (Millipore, UFC9030).

**Immunostaining with Rdots.** Cos-7, HeLa, and SKBR3 cells were purchased from ATCC. Cos-7 and HeLa cells were cultured in DMEM medium (Invitrogen, 11965) supplemented with 10% FBS (Invitrogen, 16000) and 1% penicillin–streptomycin (Invitrogen, 15140). SKBR3 cells were cultured in McCoy's 5a Medium Modified (ATCC, 30-2007) supplemented with 10% FBS (Invitrogen, 16000) and 1%

penicillin–streptomycin (Invitrogen, 15140). All cell cultures were maintained in a humidified environment at 37 °C and 5% $CO_2$. Mouse colon frozen tissue sections were purchased from Zyagen Labs (Zyagen Labs, MF-311) and stored desiccated in −80 °C as instructed by the manufacture.

**Immunostaining with cells**. Before staining, cells were seeded on 12 mm No.1 round coverslips (Thermo Scientific, 12-545-80 P) and allowed to settle overnight to reach 50% confluence. Cells were washed twice with PBS (Gibco, 14200075) and then extracted with pre-warmed extraction buffer containing 0.1 M PIPES (Sigma, P1851), 1 mM EGTA (Sigma, E3889), and 1 mM $MgCl_2$ (Sigma, M4880) for 30 s at room temperature, followed by immediate fixation with freshly made 4% PFA (Electron Microscopy Sciences, 15713) and 0.1% GA (Sigma, G7776) in PBS for 15 min. Cells were subsequently washed with PBS for 3 times to remove fixatives. Next, cells were permeabilized with 0.5% Triton-X100 for 5 min and then blocked with blocking buffer containing 5% BSA and 0.1% Triton-X100 for 0.5 h at room temperature. Primary antibodies were diluted to the desired concentrations in the staining buffer containing 2% BSA and 0.1% TritonX-100 and incubated with cells for 2 h at room temperature or overnight at 4 °C. After this, cells were washed with PBS three times, each for 5 min, and then blocking buffer for 30 min at room temperature to completely remove excessive antibodies. IgG-conjugated Rdots were then diluted in staining buffer to yield 30 nM final concentration, and incubated with cells for 2 h at room temperature or overnight at 4 °C. Following the incubation, cells were washed with PBS 3 times, 5 min each, and then blocked with blocking buffer for 15 min at room temperature, followed by a wash with PBS before imaging.

**Immunostaining of cytoskeleton with SpA captured primary antibody**. Before staining, cells were seeded on 12 mm No.1 round coverslips and allowed to settle overnight to reach 50% confluence. Cells were washed twice with PBS and then extracted with pre-warmed extraction buffer containing 0.1 M PIPES, 1 mM EGTA, and 1 mM $MgCl_2$ for 30 s at room temperature, followed by immediate fixation with freshly made 4% PFA and 0.1% GA in PBS for 15 min. Cells were subsequently washed with PBS for 3 times to remove fixatives. Next, cells were permeabilized with 0.5% Triton-X100 for 5 min and then blocked with the blocking buffer containing 5% BSA and 0.1% Triton-X100 for 0.5 h at room temperature. Primary antibodies were mixed with SpA conjugated Rdots (20 µg/ml anti-a-tubulin IgG with 1 µM Rdots) and incubated at room temperature for 1 h so that IgG could be immobilized on Rdots. Then the mixture was diluted 20x times in staining buffer containing 2% BSA and 0.1% TritonX-100 and incubated with cells for 2 h at room temperature or overnight at 4 °C. Following the incubation, cells were washed with PBS three times, 5 min each, and then blocked with blocking buffer for 15 min at room temperature, followed by a wash with PBS before imaging.

**Immunostaining with cells by primary antibody conjugated Rdots**. Before staining, SKBR3 cells were seeded in 4-well plates and allowed to settle overnight to reach 80% confluence. Cells were then detached by trypsin, washed twice with PBS, and fixed with freshly diluted 4% FPA for 15 min. Then anti-CD44 antibody (14044185, Invitrogen) conjugated Rdots were diluted to 30 nM with staining buffer containing 3% BSA. Cells were then resuspended in the staining solution and were allowed to incubate at room temperature for 30 min. Cells were thoroughly washed three times with an excessive amount of PBS. Lastly, cells were allowed to settle on poly-lysine-coated coverslips before imaging.

**Immunostaining with frozen tissue sections**. The frozen tissue section slides were first dried at room temperature for 30 min after taken out from −80 °C freezer. Then they were fixed with freshly diluted 4% PFA for 15 min, followed by three washes with PBS, each for 5 min to remove the fixative and OCT. The slides were subsequently blocked with the blocking buffer containing 5% BSA and 0.1% Triton-X100 for 0.5 h at room temperature. Primary antibodies were firstly diluted to the desired concentrations in the staining buffer containing 2% BSA and 0.1% TritonX-100 and then added to the slides to incubate for 2 h at room temperature or overnight at 4 °C in a humidified chamber. After this, slides were washed with PBS three times, each for 5 min, and then blocking buffer for 30 min at room temperature to completely remove excessive antibodies. IgG-conjugated Rdots were then diluted in staining buffer to yield 30 nM final concentration, and incubated with the slides for 2 h at room temperature or overnight at 4 °C. Following the incubation, tissue slides were washed with PBS three times, 5 min each, and then blocked with blocking buffer for 15 min at room temperature, followed by a wash with PBS before imaging.

The antibodies and their concentrations can be found in the supplementary documents (Supplementary Table 3).

**Statistics and reproducibility**. For representative images shown in Fig. 4a–d, a total of 41 biological replicates with 5 technical replicates for each biological replicate were taken as routine quality control of Rdots preparation. For representative images shown in Fig. 4e–t, 3 biological replicates with 8 technical replicates for each biological replicate were performed with consistent results. For representative images shown in Fig. 4u–w, a total of 11 biological replicates with 20

technical replicates for each biological replicate were taken as routine quality control of Rdots preparation. For representative images shown in Fig. 5, 3 biological replicates with 10 technical replicates for each biological replicate were performed with consistent results.

**Reporting summary**. Further information on research design is available in the Nature Research Reporting Summary linked to this article.

## Data availability
The data that supports this study is available from the corresponding author upon reasonable request.

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

## Acknowledgements
We thank Dr. Lixue Shi for her generous help in the project. W.M. acknowledges grant support from NIH R01 (GM128214) and NIH R01 (GM132860).

## Author contributions
Z.Z. and W.M. designed the project. Z.Z., C.C., and S.W. performed the study, S.W. contributed to compounds synthesis, T.J. acquired EM images, and all authors participated in manuscript writing.

## Competing interests
The authors declare no competing interests.
