## [Peer Review File · Nature Communications]

REVIEWER COMMENTS

Reviewer #1 (Remarks to the Author):

In this manuscript, Prof. Min and his team present the incorporation of their "Carbow" dyes into polystyrene nanoparticles at high concentrations, which enables a low-detection limit and dense multiplexing.

Overall, I thought the manuscript was clear, concise, presented strong evidence and should be accepted with no major revisions other than the following:

* Throughout the text, I think "sensitivity" is sometimes applied, possibly incorrectly, when "detection limit" is actually meant. To a degree, I realize this is an analytical chemistry/metrology vs non-scientific usage, but I'd encourage the authors to re-examine the use of "sensitivity" throughout the text.

* In figure 3a, I strongly encourage the authors to (a) continue the black regression line down to 0 M for clarity and (b) plot the confidence intervals (e.g., 95%) for the linear regression, and (c) give the theoretical standard deviation (maybe based on shot-noise and detection dark/read noise sources), example 900 pM +/- 100 pM.

These 3 things would give a much clearer view of the detection limits. I realize that the confidence intervals ~0 M may show 900 pM +/- 500 pM, but I think this technology is extremely impressive either way.

Reviewer #2 (Remarks to the Author):

The paper is overall well written, but the claims are not entirely supported. The authors try to describe the overall challenges with current techniques for immuno-staining on tissues with both current fluorescent and Raman techniques. However, I believe their interpretation of these limitations is misunderstood. They claim that current Raman spectroscopy based techniques are lacking in sensitivity and specificity. However, there are reports that show that SERS-based nanoparticle techniques are in fact ultra-sensitive and specific. Not sure what their metric for comparison is here.

There are also several claims that aren't supported or well demonstrated in the paper. Single particle detection was not demonstrated well enough within the paper. It would be nice to have a TEM/Raman image co-registered to demonstrate the ability to detect single nanoparticles. Especially with the small form factor of their 20 nm particle, the optical resolution of the microscopic imaging device won't be able to resolve single versus multiple nanoparticles in a given field of view.

Their claim about obtaining the highest sensitivity among all Raman imaging methods is also not supported as they have not done any head to head comparisons with other SERS-based methods. There is clearly a need for better immuno-staining procedures, and Raman strategies could certainly offer improved multiplexing capabilities. They mention how fluorescence is limited to 4-5 colors and that Raman can do over 100 simultaneously (please reference this), but in their paper they only show 6, and since they are relying on single peaks for identification of their individual Rdot batches in the 2000-2250 cm⁻¹ range, I am skeptical that they would be able to create many more Rdots to successfully multiplex simultaneously.

Their claims about it taking a long time to acquire SERS spectra for each pixel and the image resolution being low are also not supported by the literature, reports have shown 10-30 msec per acquisition DOI: 10.1002/jbio.201200148. It is true that SERS nanoparticles are larger in size (>50 nm), and would have a harder time to diffuse into cellular structures, however for tissue staining on a slide as they are proposing, this wouldn't be a problem, since the cells are already splayed open and antigens are accessible without diffusion. Their claims would be better supported for looking at live cell experiments.

Their claims about detection limits and single particle sensitivity imaging and comparing to SERS are unfounded. There were no studies to show that their 20 nm particles behaved any better than a larger nanoparticle in terms of cellular diffusion. Would be nice to see if live cells are more likely to take up 20 nm particles than 50 or 100 nm particles. Again, it's important to note that for tissue staining, this "size advantage" would likely not matter.

In Figure 2c, it is difficult to interpret the data. They tested 19 Carbow dyes, but it isn't clear if they dyes are all incorporated into Rdots. It looks as if they only tested 6 as RDots, but then in the text talk about how they successfully incorporated all of them into PS nanos. But then later talk about how the full panel could be used making it possible to generate more than 20 kinds of resolvable Rdots as if they hadn't been made yet.

TEM imaging of the final product would be good to demonstrate the size and shape of the nanos. 900 pM sensitivity isn't that impressive, as investigators have been able to report fM sensitivity range for SERS Nanos.

In the beginning of the paper they claim single particle detection, but then in their results section, they suggest that it might be possible but are not completely sure. The data that they used to present the trend of aggregates and single nanoparticle detection wasn't compelling. The claim needs further experimental rigor to support single nanoparticle detection. Just because they show more intensity doesn't mean that aggregates exist in those spots.

Figure 4 is busy and hard to follow. Is the scale bar 20 microns for all the images a-j?

Also figure 4 d doesn't look like negligible background, in fact it looks just like Fig 4 c but less enhanced.

The markers chosen appear to be cell surface receptors, and therefore don't support the claims that their ultra small 20 nm nanoparticle can easily diffuse in cells.

In the end the authors only really used two batches of their Rdots to demonstrate "high" multiplexing. The other two colors were fluorescent probes for their "four color imaging". This did show the ability to have good compatibility between fluorescence and Raman stains, but doesn't support the claims that Raman have an advantage over conventional fluorescence, since fluorescence can easily beat two colors. Their narrowband SRS imaging technique also has a major disadvantage over existing Raman techniques in that it requires imaging at different Raman frequencies at a time, which would add to the overall time to acquire the images. They also didn't discuss how long it took to acquire the images.

Overall, the authors have described an interesting approach to develop new ultras-small Raman nanoparticles. However, they have over claimed several things about their Raman imaging technique throughout the paper without convincing experimental data to back up their claims. They also

misrepresent the literature and what has been achievable in the past with alternative Raman imaging methods. Their technique is certainly interesting and should be further developed, but with the right experiments to support their claims. Their limitations also need to be fully understood and discussed in the paper.

Reviewer #3 (Remarks to the Author):

Occasionally one reads a paper that is at is simultaneously simple and compelling, yet will clearly have an important impact in its field. For me this was such a paper.

Zhao et al. have demonstrated that by simply swelling ~20 nm polystyrene beads in a solvent containing a relatively high Raman cross-section aromatic molecule, then de-swelling them by adding buffer, they could load the beads with sufficient density of Raman scatterers to yield an overall cross-section similar to that of a fluorescent dye. They showed that single-particle detection is possible.

They further used established techniques for functionalizing the beads and showed that, as expected the small beads could reliably label intracellular features and surface proteins.

The simplicity and likely utility of the Rdots described here are sufficiently compelling that I envision my own lab using them, perhaps extensively. The technologies and protocols are easily in the reach of any lab that is familiar with fluorescent labeling of biological materials, and could possibly spur widespread use of these probes with imaging based on spontaneous Raman scattering, as they have no bleaching problems.

I recommend that the article be published, but a couple of minor issues should be addressed first:

1) With such a high load of Raman scatterers, these Rdots will undoubtedly absorb and vibrationally dump significant amounts of heat into the system. The authors should at the very least mention this. I would recommend at least a back-of-the envelope calculation to arrive at an upper limit and a brief discussion of the likely temperature excursions generated locally during imaging. I suspect that the local heating may be significant. This may not be a problem for fixed samples, but may be a factor in live-cell imaging.

2) The authors should provide the reader with the actual concentration of the Carbow dye in the "20x Carbow dye DMSO stock solution" described under "Preparation of Rdots".

Response to Reviewer #1:

In this manuscript, Prof. Min and his team present the incorporation of their "Carbow" dyes into polystyrene nanoparticles at high concentrations, which enables a low-detection limit and dense multiplexing.

Overall, I thought the manuscript was clear, concise, presented strong evidence and should be accepted with no major revisions other than the following:

* Throughout the text, I think "sensitivity" is sometimes applied, possibly incorrectly, when "detection limit" is actually meant. To a degree, I realize this is an analytical chemistry/metrology vs non-scientific usage, but I'd encourage the authors to re-examine the use of "sensitivity" throughout the text.

* In figure 3a, I strongly encourage the authors to (a) continue the black regression line down to 0 M for clarity and (b) plot the confidence intervals (e.g., 95%) for the linear regression, and (c) give the theoretical standard deviation (maybe based on shot-noise and detection dark/read noise sources), example 900 pM +/- 100 pM.

These 3 things would give a much clearer view of the detection limits. I realize that the confidence intervals ~0 M may show 900 pM +/- 500 pM, but I think this technology is extremely impressive either way.

We greatly appreciate the reviewer's highly supportive comments and valuable suggestions.

We thank the reviewer's comments on the use of 'sensitivity' and 'detection limit'. We have re-examined the texts throughout and made changes of 'sensitivity' to 'detection limit' or 'detection sensitivity' when we are referring to the minimum detectable concentration.

Regarding Figure 3a, for suggestion (a), 5 nM was the minimal concentration we measured, and that was the reason we did not continue the regression line in the original manuscript. In the revised version, we measured the signal intensity at 0M and included the new data in the figure. The regression line is now down to 0 M, as the reviewer suggested. However, it should be noted that our imaging system has illumination volume $1\sim 2 \times 10^{-16}$ L, and the corresponding molarity of single particle is ~ 5 nM. Therefore a linear relationship may not be strictly expected below 5 nM, as we cannot guarantee a perfectly even dispersion of every individual nanoparticle. For this reason, the regression line below 5 nM is dotted.

For suggestion (b), we've modified Fig. 3a and its corresponding legend to include 95% confidence interval.

For suggestion (c), we've estimated the theoretical detection limit to be 900 ± 50 pM (mean \pm 99% confidence interval of the noise measurement), the text was modified accordingly.

Response to Reviewer #2:

The paper is overall well written, but the claims are not entirely supported. The authors try to describe the overall challenges with current techniques for immuno-staining on tissues with both current fluorescent and Raman techniques. However, I believe their interpretation of these limitations is misunderstood. They claim that current Raman spectroscopy based techniques are lacking in sensitivity and specificity. However, there are reports that show that SERS-based nanoparticle techniques are in fact ultra-sensitive and specific. Not sure what their metric for comparison is here.

We thank the reviewer's appreciation of our paper and overall highly supportive comments. We understand there are several claims that the reviewer is concerned about. As we will elaborate below, some of the concerns are because of the lack of sufficient explanation in our original article, while some are likely due to the reviewer's misunderstanding of our claims. The reviewer mainly raised three questions:

- 1) **Whether or not Rdots have single particle detection sensitivity.** To address this question, we've acquired new electron microscopy images as suggested by the reviewer, and included new analysis and discussion in the text. Together these new data and analysis further confirm single particle detectability.
- 2) **Whether or not Rdots are superior to SERS nanoparticles.** First, we admit that we did not have head-to-head experimental data to compare between SERS and Rdots of the same 20-nm size. The discussion in our original manuscript was of theoretical interest only. Second, we realize that SERS nanoparticles of the same size (~20 nm) were seldomly reported or utilized in the literature, which again makes the comparison less critical. Furthermore, it is never our main focus to claim Rdots are superior to SERS nanoparticles. All techniques have their own scope of applications. Based on these three considerations, we've removed the theoretical discussion on the sensitivity comparison with SERS from the main text to the supplementary document. Meanwhile, we've constrained our claims to be about *organic* based Raman nanoparticles.
- 3) **Whether or not the compact sizes of Rdots are important for our applications.** The reviewer might have misunderstood our data by thinking our demonstrations were only done for cell surface markers. In fact, the major part of cellular staining was designed to prove that Rdots can diffuse into cells and stain *intracellular* markers such as cytoskeleton. This is consistent with reports by other groups that smaller sizes of nanoparticles help them diffuse into cells more easily. To further elaborate on this point, we also included additional data in the revision to demonstrate that 50 nm Rdots (a few times larger than our 20-nm Rdots developed in our paper) cannot stain intracellular markers, likely because of diffusion barrier. We believe the compact size is one of the enabling aspects of Rdots.

Hence, we've performed new experiments and added new data, analysis and discussion to the revised version. We hope these can help clarify the remaining concerns and misunderstandings of the reviewer.

There are also several claims that aren't supported or well demonstrated in the paper. Single particle detection was not demonstrated well enough within the paper. It would be nice to have a TEM/Raman image co-registered to demonstrate the ability to detect single nanoparticles. Especially with the small form factor of their 20 nm particle, the optical resolution of the microscopic imaging device won't be able to resolve single versus multiple nanoparticles in a given field of view.

We thank the reviewer's suggestion of including correlative TEM/SRS images. In fact, we made several attempts along this direction, however, they were unsuccessful largely, because technical difficulties prohibited us from achieving co-registered EM/SRS images. On the sample preparation side, TEM requires the sample to be on top of a copper grid mesh, which unfortunately significantly interferes with SRS imaging which is operating on light transmission. Additionally, Rdots are better to be submerged in aqueous solutions to dissipate the heat generated by excited vibrational modes. Yet, electron microscopy requires the sample to be dried. On the imaging co-registration side, it is also very technically challenging to locate the same nanometer-sized particles from a centimeter-wide glass slide, because the dehydration and rehydration will inevitably change the spatial location of nanometer-sized particles. These technical difficulties prevent us from having co-register EM/Raman images. Indeed, this is also seldomly done in the literature.

Although we do not have the co-registered TEM/Raman images, we managed to obtain new SEM images and added to the revised version. We have several strong reasons (including SEM morphology, detailed signal to noise analysis, and signal distribution analysis) to believe that we have observed single particle events.

- 1) We have acquired new SEM images of Rdots (Response Figure 1), which confirms that the majority of Rdots are indeed monodisperse on coverslips. This is consistent with our dynamic light scattering (DLS) data (Fig. 2 d and e) that Rdots form monodisperse colloid in aqueous solution -- we only observed one major peak centered at 21 nm in the size distribution data. Such consistency makes sense because Rdots without biofunctionalization possess strong surface charges from surface carboxyl groups (zeta potential = -46 ± 12 mV at pH7.4) and the repulsive force is strong.

Response Figure 1. SEM image of Rdots2220. Scale bar: 50-nm. It can be seen from the image that majority of Rdots are single particle dispersed on coverslip with only a few forming small aggregates. The shape (donut-like) of Rdots in our SEM images is consistent with that of polystyrene nanoparticles reported in the literature (Isa, L. *et al. Nat Commun* **2**, 438 (2011)).

- 2) Detailed analysis of signal-to-noise ratio supports our conclusion of single-particle detectability. Our previous study (ref21) showed Carbow 2yne has a detection limit around 4.3 μM (when $\text{SNR} = 1$), which is equivalent to 250 ~ 500 molecules within the excitation volume. As mentioned in the manuscript, we have ~2700 Carbow 2yne dyes entrapped in each, so we would expect SNR from single Rdots2220 is around 8:1. This result is for Rdots2220 measured in solution in which the particles are monodisperse. And indeed, under SRS imaging, the SNR of ‘dimmest’ single particles candidates on coverslip is also ~8, which is consistent with the result from solution measurement.
- 3) The high SNR (8:1) of the ‘dimmest’ imaged spots shown in Fig. 3 is much greater than the detection limit ($\text{SNR}=1$). This means if there were a spot of lower intensity, we would have been able to detect it. Yet we did not find additional peaks of weaker signal in the histograms shown in Fig. 3d and e, nor in the single particle images shown in Fig. 3b and f. This strongly suggests that the ‘dimmest’ spots are indeed single particles that exhibit the elementary signal intensity.
- 4) The distribution analysis of the intensity also supports the claim of sing-particle detectability. We quantified signal intensity of hundreds of possible single particle images and plotted the signal intensity distribution (Fig. 3d and e). It’s to be noted that SRS signal is strictly linearly dependent on the number of vibrational modes. At basic condition, where even stronger repulsive force is expected due to deprotonation of surface carboxyl groups, two peaks are clearly visible in the distribution. It is worth noting that the second peak has exactly twice the intensity as the first peak. Such quantized signal distribution strongly suggests that the first peak correlates to single particle while the second peak correlates to double-particle aggregates. Similarly, in Fig. 3f-g, we visualized single Rdots and Rdots aggregates and quantified their signal intensity. We found that the intensity of each individual spots are always whole-number multiples of a lowest value, which strongly suggest an elementary signal intensity coming from single Rdots. If the minimal aggregates were to start from 2 particles for example, then the second dimmest spots consisting of 3 particles would be 3/2 of the brightness. Yet, we did not observe this in either the histogram (Fig. 3d) or the single particle line profiles (Fig. 3g). All the peaks are integer-multiple times brighter than the dimmest peaks. In fact, the same principle was behind how the elementary electric charge was historically discovered and measured in the famous Millikan oil-droplet experiment.

We hope the above evidence from both experiments and statistical analysis could help clear the confusion and convince the reviewer of sing-particle detectability. We have included much of these into the revision.

Their claim about obtaining the highest sensitivity among all Raman imaging methods is also not supported as they have not done any head to head comparisons with other SERS-based methods.

We thank the reviewer's comments, and we totally agree that SERS nanoparticles are very powerful tools in analytical chemistry for detection and analyzing a variety of biomolecules.

First, we admit that we did not have head-to-head experimental data to compare between SERS and Rdots of the same 20 nm size. The discussion in our original manuscript was of theoretical interest only. Second, we realize that SERS nanoparticles of the same size (~20 nm) were seldomly reported or utilized in the literature, which again makes the comparison less critical. Furthermore, it is never our main focus to claim Rdots are superior to SERS nanoparticles. All techniques have their own scope of applications. Based on these three considerations, we've removed the theoretical discussion on the sensitivity comparison with SERS from the main text to the supplementary document. Meanwhile, we've constrained the claims to be about organic based Raman nanoparticles.

There is clearly a need for better immuno-staining procedures, and Raman strategies could certainly offer improved multiplexing capabilities. They mention how fluorescence is limited to 4-5 colors and that Raman can do over 100 simultaneously (please reference this), but in their paper they only show 6, and since they are relying on single peaks for identification of their individual Rdot batches in the 2000-2250 cm^{-1} range, I am skeptical that they would be able to create many more Rdots to successfully multiplex simultaneously.

The theoretical 100-color was calculated based on 10 cm^{-1} minimal linewidth that we can resolve on our set-up and the 1000 cm^{-1} span in the cell silent region (ranging from 1800 to 2800 cm^{-1}). In this article, we primarily used alkyne and nitrile as the reporting group, however, we're working on other vibrational moieties (such as thiols or conjugated alkyne and nitrile) which will hold the promise for fully taking advantage of the entire silent region. We have now emphasized in the revision that this 100-color is "theoretical".

In this study, we've generated 25 Rdots in total successfully, 6 of which were with Carbow dyes, and 19 of which were with commonly used Raman probes containing alkyne or nitrile group. Most of these 19 common Raman probes are commercially available and the other few of them can be easily synthesized. We chose these 19 common Raman probes to show that generalization of the method we developed to prepared Rdots, as not all labs are accessible to Carbow dyes. We believe the successful generation of Rdots with these molecules would make Rdots accessible to many labs.

To clear the potential confusion, we've added a table as Table S2 containing all the molecular structures of these 19 Raman probes and 6 Carbow dyes used in the article. Additionally, we also added another supplementary figure Fig. S4 (also Response Figure 2 below) to show that at least 10 Rdots with resolvable Raman peaks could be used.

Response Figure 2. 10 Rdots with resolvable Raman peaks. Total 10 Rdots with resolvable Raman peaks were generated with selected Carbow dyes and commonly used Raman probes.

Their claims about it taking a long time to acquire SERS spectra for each pixel and the image resolution being low are also not supported by the literature, reports have shown 10–30 msec per acquisition DOI: 10.1002/jbio.201200148. It is true that SERS nanoparticles are larger in size (>50 nm), and would have a harder time to diffuse into cellular structures, however for tissue staining on a slide as they are proposing, this wouldn't be a problem, since the cells are already splayed open and antigens are accessible without diffusion. Their claims would be better supported for looking at live cell experiments.

We agree that acquisition speed is one of the key aspects of any imaging tool, as it will not be practically useful if the acquisition time is prohibitively long. We appreciate the valuable reference the reviewer provided.

However, the reviewer might have misunderstood the literature which actually supports that Rdots combined with SRS is a fast imaging tool. In the article the reviewer referred to states that the ‘Rapid immunoSERS microscopy’ takes ‘30 msec acquisition time per pixel’. For imaging with Rdots as shown in our original paper, we used 10 ~ 30 us pixel dwell time / time constant (which translates to ~8 sec per 512x512 image), as mentioned in the Methods section. This means that doing vibrational imaging with Rdots coupled with SRS is at least 1000 times quicker. The quick imaging speed enables us to image a large area of tissue. For example, the imaging of whole cross section of mouse colon slice (~2 x 2 mm) with diffraction limited spatial resolution can be done within 10 minutes (Fig. 4v).

We’ve added the suggested citation as ref14. We also added the following sentences to the manuscript (section *Rdots enable immunostaining in cells and tissues*) discussing the time it took to take SRS images with Rdots:

‘For SRS imaging, we achieved satisfactory quality with 10 ~ 30 us time constant and the same pixel dwell time, and it took less than 8 s to acquire an image of 512 by 512 pixels.’

Admittedly, for our system that operates on narrow band excitation, we only acquire 1 channel at a time, which means that it would take 8*10 s plus laser tuning time after each channel (usually around 15 s) for 10 color imaging. Hence, even for 10 color, the narrow band excitation method is still 100 times faster than Rapid immune-SERS. Furthermore, Rdots are also compatible with hyperspectral SRS imaging systems for truly simultaneous excitation and detection of multiple channels. For example, the system shown in Response Figure 3 below was reported to be able to acquire 16 SRS channels simultaneously (Liao, C. *et al. Light Sci Appl* **4**, e265 (2015)). When coupling Rdots with such hyperspectral SRS system, we expect the imaging speed can be nearly 1000 times faster than Rapid immune-SERS. We have now included this reference in our revised version to comment on the future prospects.

Response Figure 3: A lab-built hyperspectral SRS microscope by broadband excitation and parallel detection. (The figure is from Liao, C. *et al. Light Sci Appl* **4**, e265 (2015))

We believe that the small size of nanoparticles is actually critical for successful immunostaining in cells. In fact, the major part of cellular staining in our manuscript was designed to prove that Rdots can diffuse into cells and stain intracellular markers such as cytoskeleton. This is consistent with reports by other groups that smaller sizes of nanoparticles help them diffuse into cells more easily. We also include additional experimental data to further demonstrate larger 50 nm Rdots cannot stain intracellular markers, likely because of diffusion barrier (Response Figure 4 below). The compact size is one of the enabling aspects of Rdots, and explains the lack of immunostaining of intracellular markers by larger SERS nanoparticles in the literature.

Their claims about detection limits and single particle sensitivity imaging and comparing to SERS are unfounded. There were no studies to show that their 20 nm particles behaved any better than a larger nanoparticle in terms of cellular diffusion. Would be nice to see if live cells are more likely to take up 20 nm particles than 50 or 100 nm particles. Again, it's important to note that for tissue staining, this "size advantage" would likely not matter.

We agree that one of the pitfalls of using the nanoparticles as staining reagents is the difficulty in cellular diffusion (a nice review article discussing this issue: Jin, D. et al. *Nat Methods*, 2018, cited as ref4 in the submitted manuscript). In the original manuscript, we actually discussed this problem in the first paragraph of the Results section. Many studies especially in the field of quantum dots have investigated this problem. For example, Howarth et al. (Howarth, M. et al. *Nat Methods*, 2008, cited as ref23) found Qdots of reduced size improved labeling of glutamate receptors. Liu et al. (Liu, Y. et al. *Nat Commun*, 2018, cited as ref24) found that Qdots of reduced size improved intracellular mRNA labeling. Baum et al. (Baum, M., et al. *Nat Commun*, 2014, cited as ref22) also found that the interior of cells appeared as porous media with a throat size of ~20 nm.

For these reasons, we aimed to develop an ideal material whose hydrodynamic diameter should be comparable to or less than 20 nm. We actually tested nanoparticles of 50 nm diameter to stain the same α -tubulin target as shown in Fig. 4. Indeed, the larger 50 nm Rdots had difficulties in diffusing into cells and showed poor immunostaining results (Response Figure 4).

Response Figure 4. Immunostaining of α -tubulin with 50 nm Rdots2220 in cos7 cell. Imaging of C-H stretching mode (2940 cm^{-1}) confirms the existence of the cell. No SRS signal or filament structure of microtubule stained by 50 nm Rdots2220 is observed (2220 cm^{-1}). Scale bar: 20 μm .

As for immunostaining with tissue slices, we agree with the reviewer that nanoparticles have less difficulties in terms of diffusion. We actually explicitly said the same thing as the reviewer suggested in the 3rd paragraph in the Discussion and Conclusion section.

In Figure 2c, it is difficult to interpret the data. They tested 19 Carbow dyes, but it isn't clear if they dyes are all incorporated into Rdots. It looks as if they only tested 6 as RDots, but then in the text talk about how they successfully incorporated all of them into PS nanos. But then later talk about how the full panel could be used making it possible to generate more than 20 kinds of resolvable Rdots as if they hadn't been made yet.

We thank the review's question regarding the Raman dyes that were used. There might be some confusions here. We've generated 25 Rdots in total successfully, 6 of which were with Carbow dyes, and 19 of which were with commonly used Raman probes containing alkyne or nitrile group. We chose these 19 common Raman probes to show that generalization of the method we developed to prepared Rdots, as not all labs are accessible to Carbow dyes. Most of these 19 common Raman probes are commercially available and the other few of them can be easily synthesized. We believe the successful generation of Rdots with these molecules would make Rdots accessible to almost all labs.

To clear the confusion, we've added a table as Table S2 containing all the molecular structures of these 19 Raman probes and 6 Carbow dyes used in the article. Additionally, we also added another supplementary figure Fig. S4 (also Response Figure 2) to show that at least 10 Rdots with resolvable Raman peaks could be used.

TEM imaging of the final product would be good to demonstrate the size and shape of the nanos.

We thank the reviewer for the suggestion. Although TEM images are great characterization of the morphology of Rdots, the TEM facility in our university is currently affected by the pandemic. Instead, we've included SEM images to the supplementary document (newly added Fig. S3 in our revision). As shown in the images, we found no significant size or morphology changes of the nanoparticles before and after the Raman probe doping procedure. This is also consistent with our DLS measurement.

900 pM sensitivity isn't that impressive, as investigators have been able to report fM sensitivity range for SERS Nanos.

We totally agree with the reviewer that SERS nanoparticles are powerful and sensitive tools in analytical chemistry. Even the detection of single molecule has been demonstrated with SERS NPs (ref37, SI ref3~6).

However, the sub-nM detection limit discussed in our original manuscript was referring to the concentration of Rdots themselves, but not their analytes. We think what the reviewer referred to as fM sensitivity for SERS is the detection limit of the analytes. For example, Yang et al.

reported aM detection limit (Yang, S. *et al. Proc National Acad Sci* **113**, 268–273 (2016).), but the low aM concentration was referring to the analyte (Rhodamine 6G in this case), not the concentration of nanoparticles themselves.

For our setup, which has illumination volume $1\sim 2 \times 10^{-16}$ L, the corresponding molarity of single particle is on the order of 5 nM. Detection of nanoparticle themselves significantly below this level is essentially the same as detection of single nanoparticles. That is why single particle sensitivity is regarded as the ultimate sensitivity. In this sense, we've demonstrated single particle sensitivity in our study.

In the beginning of the paper they claim single particle detection, but then in their results section, they suggest that it might be possible but are not completely sure. The data that they used to present the trend of aggregates and single nanoparticle detection wasn't compelling. The claim needs further experimental rigor to support single nanoparticle detection. Just because they show more intensity doesn't mean that aggregates exist in those spots.

The reviewer might be referring to the following sentence when the reviewer states we were not completely sure:

'However, unlike single molecule fluorescence whose stoichiometry can be determined by observing the abrupt photobleaching steps from individual fluorophores, it is difficult to confirm the signal from diffraction limited spots in the SRS image was indeed from single particles or from aggregation of a few single particles.'

We might be unclear in the sentence above, but what we really meant was that it's difficult to confirm single particles with the commonly used photobleaching methods. That's why we adopted another method to make sure single-particle detection limit. We've modified the manuscript to make this point clearer.

Although we do not have the co-registered TEM/Raman images, we managed to obtain new SEM images and added it to the revised version. We have several reasons (SEM morphology, signal to noise analysis and signal distribution analysis) to believe that we have observed single particle events.

- 1) We acquired new SEM images of Rdots (Response Figure 1), which confirms that the majority of Rdots are monodisperse on coverslip. This is consistent with our DLS data (Fig. 2 d and e) that Rdots form monodisperse colloid in aqueous solution -- we only observed one major peak centered at 21 nm in the size distribution data, indicating the majority of the Rdots are dispersed as single particles. This makes sense because Rdots without biofunctionalization process strong surface charge from surface carboxyl groups (zeta potential = -46 ± 12 mV at pH7.4) and the repulsive force is strong.
- 2) Our analysis of signal to noise ratio supports the claim of single-particle detectability. Our previous study (ref21) showed Carbow 2yne has a detection limit (when SNR = 1) around 4.3 μ M, which is equivalent to 250 ~ 500 molecules within the excitation volume. As mentioned in the manuscript, we have ~2700 Carbow 2yne dyes entrapped in each

Rdots2220, as measured in bulk solution (in which the particles are monodisperse). So we would expect SNR from single Rdots2220 is around 8. Indeed, under SRS imaging, the SNR of 'dimpest' single particles candidates is ~ 8 , which is consistent with our solution estimation. On the other hand, the high SNR of the 'dimpest' imaged spots shown in Fig. 3 is much greater than the detection limit. This means if there were a spot of lower intensity, we would have been able to detect it. Yet we did not find additional peaks of weaker signal in histograms shown in Fig. 3d and e, nor in the single particle images shown in Fig. 3b and f. This suggests that the 'dimpest' spots are indeed single particles that have elementary signal intensity.

- 3) Our distribution analysis of the intensity also supports the claim of sing-particle detectability. We quantified signal intensity of hundreds of possible single particle images and plotted the signal intensity distribution (Fig. 3d and e). It's to be noted that SRS signal is linearly dependent on concentration. At basic condition, where even stronger repulsive force is expected due to deprotonation of surface carboxyl groups, two peaks are clearly visible in the distribution. It is worth noting that the second peak has exactly twice the intensity as the first peak, indicating quantized signal distribution. Such quantized signal distribution strongly suggests that the first peak correlates to single particle while the second peak correlates to double-particle aggregates. Similarly, in Fig. 3f-g, we visualized single Rdots and Rdots aggregates and quantified their signal intensity. We found that the intensity of each individual spots are always whole-number multiples of a lowest value, which strongly suggest an elementary signal intensity coming from single Rdots. If the minimal aggregates were to start from 2 particles for example, then the second dimpest spots consisting of 3 particles would be $3/2$ of the brightness. Yet, we did not observe this in either the histogram (Fig. 3d) or the single particle line profiles (Fig. 3g). All the peaks are integer-multiple times brighter than the dimpest peaks. In fact, the same principle was underlying how elementary electric charge was discovered and measured in the famous Millikan oil-droplet experiment.

We hope the above evidence could clear the confusion and convince the reviewer.

Figure 4 is busy and hard to follow. Is the scale bar 20 microns for all the images a-j?

The scale bar for figure4 a~j is 20 μm for all the images. We've added the length of the scale bar of Fig. 4 c and d in the legend.

Also figure 4 d doesn't look like negligible background, in fact it looks just like Fig 4 c but less enhanced.

The intensity scale (i.e., the brightness lookup table) for Fig. 4c and Fig. 4d are identical in the original manuscript. The negligible 'shade' appearing in Fig. 4d is due to minor cross-phase background, usually a result from slightly misaligned optics. Nevertheless, we thank the reviewer for pointing this out. We have realigned the optics in the lab and acquired new images. We have modified Fig. 4c and Fig. 4d to reflect these changes. Note that the small dots appearing

in Fig. 4d are due to highly scattering lipid droplets which are known to leave residual cross-phase background in SRS images.

The markers chosen appear to be cell surface receptors, and therefore don't support the claims that their ultra small 20 nm nanoparticle can easily diffuse in cells.

We very much agree that unobstructed diffusion into the cells is one of the key characters of any probes aiming for immunostaining. However, the reviewer might have misunderstood our data by thinking our demonstrations were only done for cell surface markers. In fact, the major part of cellular staining in our manuscript was designed to prove that Rdots can diffuse into cells and stain intracellular markers such as cytoskeleton. Cytoskeleton is a complex intracellular network of interlinking protein filaments present in the cytoplasm of all cells. These markers localized inside the cell. Due to its rich structural features, the cytoskeleton network is commonly used to validate new staining probes.

To our best knowledge, our study is the first time that the cytoskeleton network is labeled and visualized with any kinds of Raman nanoparticles. The successful visualization and validation of the intracellular network with Rdots indicate that the small nanoparticle size is beneficial for easier diffusion. Additionally, in Response Figure 4 as mentioned earlier, we've shown that 50 nm Rdots targeting the same intracellular marker have poor staining results due to their larger size than the desirable 20nm. We hope these pieces of evidence will convince the reviewer that the small size is beneficial for cellular imaging.

In the end the authors only really used two batches of their Rdots to demonstrate "high" multiplexing. The other two colors were fluorescent probes for their "four color imaging". This did show the ability to have good compatibility between fluorescence and Raman stains, but doesn't support the claims that Raman have an advantage over conventional fluorescence, since fluorescence can easily beat two colors. Their narrowband SRS imaging technique also has a major disadvantage over existing Raman techniques in that it requires imaging at different Raman frequencies at a time, which would add to the overall time to acquire the images. They also didn't discuss how long it took to acquire the images.

We thank the critics the reviewer raised.

In our previous studies (ref21 and ref31) we already showed that 10 color simultaneous imaging with Raman dyes of similar spectral linewidth. In this article, we focused on the development of a new kind of Raman nanoparticles that can be both bright and small. Our developed Rdots, whose spectral linewidth is the same but brightness is over 1000x greater than previously reported Raman dyes, are expected to image with similar degree of multiplexing.

To further convince the reviewer of our multi-channel imaging capability, we've also added a new demonstration (Fig. S7 and Response Figure 5). The results show minimal crosstalk between channels.

Rdots2220
Rdots2195
Rdots2177
Rdots2154
Rdots2134
Rdots2093

Response Figure 5. Multiplexed optical imaging with Rdots. Lives HeLa cells were singularly stained with 1 of 5 Rdots first and then pooled for optical imaging. Rdots2154 was intentionally left out to show crosstalk. Rdots used here were not functionalized with PEG chains and processed net negative surface charge and were therefore used to stain cell surface by non-specific binding. Each individual cell can be identified unequivocally from the mixture in one 6-channel image while no signal shows up in Rdots2154 channel, indicating minimal crosstalk between channels. Scale bar: 20 μm .

We've added the following sentences to the manuscript (section *Demonstration of multiplexed cytoskeleton imaging with Rdots*) to introduce the new demonstration:

'In addition, we also achieved 6-color SRS imaging with live cell surface staining (Fig. S7). The results showed minimal crosstalk even between channels that were spectrally separated by only 18cm-1 (center wavenumber to center wavenumber), suggesting great multiplexing capacity.'

We're currently working on tissue immunostaining of 20-color, which involves detailed biological systems and questions. This is beyond the scope of the current study, which is focusing on the technical development of Rdots.

As for the time to take the images, we've added the following sentences to the manuscript (section *Rdots enable immunostaining in cells and tissues*) discussing the time it took to take SRS images with Rdots:

'For SRS imaging, we achieved satisfactory quality with 10 ~ 30 us time constant and the same pixel dwell time, and it took less than 8 s to acquire an image of 512 by 512 pixels.'

For large tissue samples such as the one shown in Fig. 4v, the entire 3 mm by 3 mm tissue slide was imaged within 10 min with diffraction limited resolution.

Admittedly, for our system that operates on narrow band excitation, we only acquire 1 channel at a time. So for 10 color imaging, it would take 8×10 s plus laser tuning time (usually around 15 s), which is still 100 times faster than Rapid immune-SERS. Furthermore, Rdots are compatible

with hyperspectral SRS systems. Take the one shown in Response Figure 3 as an example, it was reported to be able to acquire 16 SRS channels simultaneously (Liao, C. *et al. Light Sci Appl* **4**, e265 (2015)). Therefore, future employment of hyperspectral SRS techniques can take multiple channels at the same time and thus shorten the overall acquisition time.

Overall, the authors have described an interesting approach to develop new ultrasmall Raman nanoparticles. However, they have over claimed several things about their Raman imaging technique throughout the paper without convincing experimental data to back up their claims. They also misrepresent the literature and what has been achievable in the past with alternative Raman imaging methods. Their technique is certainly interesting and should be further developed, but with the right experiments to support their claims. Their limitations also need to be fully understood and discussed in the paper.

Again we thank the reviewer for his/her insight comments and many helpful suggestions, which led us to revise the manuscript in a constructive way. We believe the criticism have been fully addressed by our new data, analysis, clarification and citation to the previous studies. We hope we've convinced the reviewer that Rdots are indeed a probe of great potential to be applied in a wide range of biological studies.

Response to Reviewer #3:

Occasionally one reads a paper that is at is simultaneously simple and compelling, yet will clearly have an important impact in its field. For me this was such a paper.

Zhao et al. have demonstrated that by simply swelling ~20 nm polystyrene beads in a solvent containing a relatively high Raman cross-section aromatic molecule, then de-swelling them by adding buffer, they could load the beads with sufficient density of Raman scatterers to yield an overall cross-section similar to that of a fluorescent dye. They showed that single-particle detection is possible.

They further used established techniques for functionalizing the beads and showed that, as expected the small beads could reliably label intracellular features and surface proteins.

The simplicity and likely utility of the Rdots described here are sufficiently compelling that I envision my own lab using them, perhaps extensively. The technologies and protocols are easily in the reach of any lab that is familiar with fluorescent labeling of biological materials, and could possibly spur widespread use of these probes with imaging based on spontaneous Raman scattering, as they have no bleaching problems.

We appreciate the reviewer's highly positive and supportive comments on the development of Rdots. We too believe that the simple preparation and versatile use of Rdots would enable many possibilities in a variety of applications.

I recommend that the article be published, but a couple of minor issues should be addressed first:

1) With such a high load of Raman scatterers, these Rdots will undoubtedly absorb and vibrationally dump significant amounts of heat into the system. The authors should at the very least mention this. I would recommend at least a back-of-the envelope calculation to arrive at an upper limit and a brief discussion of the likely temperature excursions generated locally during imaging. I suspect that the local heating may be significant. This may not be a problem for fixed samples, but may be a factor in live-cell imaging.

We appreciate the reviewer's valuable suggestions. We did observe minor burning effect when the Rdots were dispersed in a poor heat dissipator such as 80% glycerol (Response Figure 6). However, we did not observe obvious heating effect during our actual imaging. For example, in Fig. 3c we have scanned the single Rdots particles for more than 10 frames with rather strong laser power density and long pixel dwell time, and we did not observe burning of Rdots. There may be two major reasons. First, Rdots were small nanoparticles, which means the surface to volume ratio is high so that the heat generated can be efficiently dissipated. Second, in all the data shown in the manuscript, we made sure that Rdots were dispersed in aqueous solution which has large specific heat capacity that could dissipate the heat quickly.

Response Figure 6. Thermo stability of Rdots in 80% glycerol. Although Rdots are stable with strong laser power in aqueous solutions which are good heat dissipator, when Rdots were dispensed in a poor thermo dissipator such as glycerol at high concentration, sample burning was observed (indicated by the arrows) after a few frames of scanning. Two factors may contribute to the thermo stability in aqueous solutions. First, Rdots are relatively small nanoparticles, and therefore they have large surface-to-volume ratio, which mean heat can be transferred quickly to the surrounding environment. Second, in the applications demonstrated in this study, we always dispensed Rdots in aqueous solutions, whose heat capacity is great. As the result, the local heating could be quickly dissipated. Scale bar: 20 μm .

We added the following sentences to the article in the 5th paragraph of Discussion and Conclusion section:

‘Another artifact might be the sample burning due to local heating since vibrational mode of a large number of molecules inside Rdots are actively excited. However, we have not observed sample burning of Rdots during our imaging experiment. There may be two major reasons. First, Rdots are small nanoparticles, whose surface to volume ratio is high so that the heat generated can be efficiently dissipated. Second, water has large specific heat capacity that dissipates the heat quickly. Since all the proposed applications require samples to be submerged in aqueous solutions, the sample burning is less likely a concern.’

2) The authors should provide the reader with the actual concentration of the Carbow dye in the "20x Carbow dye DMSO stock solution" described under "Preparation of Rdots".

We thank the reviewer’s suggestion. In our revision, a table of the concentration of dyes have been included in the latest version of supplementary information as Table S3.

REVIEWER COMMENTS

Reviewer #1 (Remarks to the Author):

You have adequately addressed all of my concerns and I commend you on your work. I support this manuscript publication.

Reviewer #2 (Remarks to the Author):

Overall the authors have addressed some of my concerns. The concerns they were unable to address with experimental data they attempted to address with theory, which is appreciated, but not suitable when making such grand claims.

The biggest concern I have is that the biggest claims they are making in the paper have still not been experimentally supported. There is a lot of justification through "theory" here. Especially with regards to the following:

1. Ultra high multiplexing capabilities "theoretically" up to 100. This again misleads the reader, just because you can estimate that 100 would be the limit you could theoretically resolve, doesn't mean that you necessarily have the ability to physically create these NPs that exhibit these precise peaks at 10 wavenumber increments.

2. Single particle sensitivity, which was inferred through several experiments but not convincingly demonstrated via microscopy. They claim to have several "strong reasons" to believe that they are achieving single particle detection, but there are still a lot of assumptions being made here, like that their system is capable of seeing a single nanoparticle. I understand their linear thought process, but what about if what their system's actual limit of detection is a cluster of two or three NPs instead of a single NP and then it scales linearly up from there. In Figure 3f, there does seem to be less intense looking Raman signals coming from neighboring spots from area 1. What is there explanation for that? They still have not been able to confirm via any other validation means that those were in fact clusters of 1, 2, 3, 4, etc... NPs.

3. "The best detection limit among all Raman-based imaging methods". I simply cannot agree with this statement.

I appreciate their attempt to make "theoretical" comparisons between SERS and their technique, but without experimental evidence, they should not be able to make such strong claims. They added a whole section on the theoretical comparisons between SERS nanoparticles and Rdots, which is appreciated and will be helpful to the reader, but does not justify their claim of being the most sensitive among all Raman-based imaging methods.

I have a problem with this statement below. It is not supported in the manuscript. The authors said they were only going to compare against other organic Raman active NPs of the same size and then make such a bold claim in the following sentence in the Abstract. Please remove this claim throughout the paper for publication.

"We demonstrated multiplexed immunostaining of specific protein targets with Rdots in mammalian cells and tissue slices, and obtained arguably the best detection limit among all Raman-based imaging methods."

Separately, this statement is also very misleading as the authors only demonstrated 2 specific protein targets with their Rdots. In the end the authors only really used two batches of their Rdots to demonstrate "high" multiplexing. The other two colors were fluorescent probes for their "four color imaging". Again, this doesn't support the claims that Raman has an advantage over conventional fluorescence, since fluorescence can easily beat two colors. Their narrowband SRS imaging technique also has a major disadvantage over conventional fluorescence microscopy imaging. Their attempt to address this using 6 different Rdots on Hela cells did not demonstrate

immunostaining of specific protein targets with their Rdots, instead it just showed what they already published previously, that they are able to deconvolve and unmix their 6 Rdots. What's really missing from this paper is the biological demonstration of their claims to be useful as a multiplexed immunostaining method in cell studies.

For the following please modify:

Indeed, this is nearly two to three orders of magnitude more sensitive than those from previously reported Raman probes^{21, 31}. Please change to "previously reported organic Raman probes"

"SRS imaging of Rdots offers arguably higher sensitivity of immunostaining than all other Raman-based methods, based on several lines of evidence."

The authors continue to make claims about having the highest sensitivity among all other Raman techniques which is not supported in any way in the manuscript. The authors even claim that they were not able to make a direct comparison to other SERS based techniques, therefore they resorted to a theoretical comparison that make a lot of assumptions in their favor.

The authors don't need to make such grand claims in order to show that they have done important work here, it's distracting.

Reviewer #3 (Remarks to the Author):

The authors have addressed my minor concerns for this very exciting article. I recommend that it be published as is.

Response to reviewer 2

Overall the authors have addressed some of my concerns. The concerns they were unable to address with experimental data they attempted to address with theory, which is appreciated, but not suitable when making such grand claims.

We are glad to know that the reviewer is satisfied with most clarification in our previous response. For his/her remaining concerns, we believe our last revision to the manuscript and the response below shall be able to address them thoroughly.

The biggest concern I have is that the biggest claims they are making in the paper have still not been experimentally supported. There is a lot of justification through “theory” here. Especially with regards to the following:

1. Ultra high multiplexing capabilities “theoretically” up to 100. This again misleads the reader, just because you can estimate that 100 would be the limit you could theoretically resolve, doesn't mean that you necessarily have the ability to physically create these NPs that exhibit these precise peaks at 10 wavenumber increments.

We've modified the theoretical 100 colors to 20 in our second revision. These 20 Raman-active probes have already been synthesized and reported in the literature.

2. Single particle sensitivity, which was inferred through several experiments but not convincingly demonstrated via microscopy. They claim to have several “strong reasons” to believe that they are achieving single particle detection, but there are still a lot of assumptions being made here, like that their system is capable of seeing a single nanoparticle. I understand their linear thought process, but what about if what their system's actual limit of detection is a cluster of two or three NPs instead of a single NP and then it scales linearly up from there. In Figure 3f, there does seem to be less intense looking Raman signals coming from neighboring spots from area 1. What is there explanation for that? They still have not been able to confirm via any other validation means that those were in fact clusters of 1, 2, 3, 4, etc... NPs.

We appreciate the review's comments. We believe our clarification below and the new change made to the manuscript shall address this remaining concern of the reviewer.

As the reviewer suggested, if our system's actual limit of detection is a cluster of two or three NPs instead of a single NP and then it scales linearly up from there, we would have seen the second intense peak in the histogram to be a non-integer multiple of the first peak. For example, if our detection limit is a cluster of two NPs having SRS intensity of 100 arbitrary unit (AU), then the cluster of three NPs would have SRS intensity of 150 AU, 1.5 times more than two-NP cluster ($100 \div 2 \times 3$). Similarly, if the detection limit is a cluster of three NPs having SRS intensity of 100 AU, then the cluster of four NPs would have SRS intensity of 133 AU ($100 \div 3 \times 4$), or 1.33 times more than three-NP cluster. However, we do not observe these non-integer multiples (such as 1.5 or 1.33) in either the intensity histogram or single NP intensity line profiles. Besides, as we stated in the manuscript, the dimmest spots we imaged in the field of view have signal-to-noise around 8. So if there existed dimmer spots made of clusters of fewer

Rdots, we would have been able to detect them with our microscope. The reviewer's concern would have been valid if our signal-to-noise of the dimmest spots were only around 1.

The line profiles of 'the less intense looking Raman signals coming from neighboring spots from area 1 in Figure 3f' are now shown in the Response Figure 1 below. The less intense spot 1' that the reviewer was referring to is only around 10% less intense compared to the neighboring spot 1, which is single particle, and about half intense than spot 2, which is a double particle cluster. The small variance between spot 1 and spot 1' is likely due to the fact that not all Rdots imaged were on the exact same z-focal plane, since they were immobilized in the agarose gel that were thicker than the depth of field which was around 1 micron.

Response Figure 1.

Therefore, we believe our experimental data strongly suggest single particle imaging of Rdots by SRS. Nevertheless, to address the reviewer's remaining uncertainty, we have softened this technical claim throughout our 2nd revision. In the newly revised manuscript, we are now stating "we have obtained evidence supporting single particle imaging" instead of "we have confirmed single particle imaging" in the Abstract, Introduction, Sub-section title, Figure caption and Conclusion. We believe that it is important to publish the experimental evidences in this paper. It is up to other researchers in the field to evaluate and repeat the experiment to reach a consensus. We hope the reviewer will be satisfied with this.

3. "The best detection limit among all Raman-based imaging methods". I simply cannot agree with this statement.

I appreciate their attempt to make "theoretical" comparisons between SERS and their technique, but without experimental evidence, they should not be able to make such strong claims. They added a whole section on the theoretical comparisons between SERS nanoparticles and Rdots, which is appreciated and will be helpful to the reader, but does not justify their claim of being the most sensitive among all Raman-based imaging methods.

I have a problem with this statement below. It is not supported in the manuscript. The authors said they were only going to compare against other organic Raman active NPs of the same size and then make such a bold claim in the following sentence in the Abstract. Please remove this claim throughout the paper for publication.

"We demonstrated multiplexed immunostaining of specific protein targets with Rdots in

mammalian cells and tissue slices, and obtained arguably the best detection limit among all Raman-based imaging methods."

Following the reviewer's comment, we have removed all such claims in the newly revised manuscript. For example, we have removed the comparison between SERS and Rdots all together from the main text as well as in the Supporting Information. After all, it is not the main point of our paper. In addition, we have limited our comparison between Rdots with organic probes only.

Separately, this statement is also very misleading as the authors only demonstrated 2 specific protein targets with their Rdots. In the end the authors only really used two batches of their Rdots to demonstrate "high" multiplexing. The other two colors were fluorescent probes for their "four color imaging". Again, this doesn't support the claims that Raman has an advantage over conventional fluorescence, since fluorescence can easily beat two colors. Their narrowband SRS imaging technique also has a major disadvantage over conventional fluorescence microscopy imaging. Their attempt to address this using 6 different Rdots on HeLa cells did not demonstrate immunostaining of specific protein targets with their Rdots, instead it just showed what they already published previously, that they are able to deconvolve and unmix their 6 Rdots. What's really missing from this paper is the biological demonstration of their claims to be useful as a multiplexed immunostaining method in cell studies.

We appreciated the reviewer's comments. We believe our clarification below and a set of new changes made to the manuscript shall address this remaining concern of the reviewer.

The concept of using Raman probes for 'multiplexed imaging' actually was first raised in the SERS field. In a 2009 SERS paper titled *Multiplexed imaging of surface enhanced Raman scattering nanotags in living mice using noninvasive Raman spectroscopy*, it quotes '*SERS nanoparticles have narrower spectral peaks, which leads to little spectral overlap—unlike quantum dots, the emission spectra of which are broad in comparison.*' (Zavaleta, C.L. et al. *Proc National Acad Sci* **106**, 13511–13516 (2009)). Clearly Raman approaches are advantageous over fluorescence approaches in terms of multiplexed imaging due to the narrow Raman peaks, and this claim has been well demonstrated in numerous papers. In this study, we have shown that our Rdots have the same narrow Raman spectral features as other Raman probes/nanoparticles have, including SERS. Since using Raman probes for imaging has been demonstrated in many previous studies cited throughout the manuscript, we believe that we do not need to further prove this already well demonstrated point.

Instead, we recognized that one of the key bottleneck of the multiplex Raman imaging (especially for immunostaining) technique is the probe itself. Therefore, instead of claiming to solve the entire multiplex Raman imaging problem completely (which clearly would be too much to claim by a single paper), we wish to focus our current paper on the critical probe development and characterization and leave its broad applications in the following-up papers targeted for biological audience. For this reason, we modified the subtitle in the multiplexed imaging section to '**Demonstration of the potential of Rdots for multiplexed imaging**'. We believe that this somewhat "scope narrowing" does not take away the main importance of our

current work, as developing new bright and compact Raman probes is at the core of multiplex Raman imaging technology.

Accordingly, we have modified the Introduction and Conclusion of the newly revised manuscript so that (1) it is clear that we will be focusing on probe development and characterization (including sensitivity study and immunostaining performance evaluation) in this paper rather than solving the entire multiplex Raman imaging completely, and (2) we explicitly state that we are demonstrating the potential of Rdots for multiplex imaging.

To acknowledge the pioneering efforts in the SERS field to raise the concept of multiplexed imaging, we added the following paragraph in the multiplexed imaging section. Interesting to note that many SERS publications also demonstrate 2-4 channel for multiplexed imaging.

“Due to the narrow Raman spectral features, multiple Raman probes with different Raman frequencies can be easily separated, resulting in high multiplexing capabilities. Indeed, multiplexed imaging has been well demonstrated by various Raman-based methods, including SERS^{38, 39}. Two to four channel imaging has been simultaneously detected with SERS measurements³⁹⁻⁴¹. Given that Rdots inherit the same narrow Raman spectral features as other Raman-based probes and that SRS imaging of Rdots is compatible with additional fluorescence channels, we sought to demonstrate the potential of multiplexed imaging with Rdots.”

For the following please modify:

Indeed, this is nearly two to three orders of magnitude more sensitive than those from previously reported Raman probes^{21, 31}. Please change to “previously reported organic Raman probes”

We have modified the manuscript as requested.

“SRS imaging of Rdots offers arguably higher sensitivity of immunostaining than all other Raman-based methods, based on several lines of evidence.”

The authors continue to make claims about having the highest sensitivity among all other Raman techniques which is not supported in any way in the manuscript. The authors even claim that they were not able to make a direct comparison to other SERS based techniques, therefore they resorted to a theoretical comparison that make a lot of assumptions in their favor.

Following the reviewer’s suggestion, we have removed all comparison with SERS, and limited our comparison with organic probes only throughout the revised manuscript.

The authors don’t need to make such grand claims in order to show that they have done important work here, it’s distracting.

We thank the reviewer’s appreciation of the importance of our work. The reviewer’s comments, both in the previous round and in the current round, have helped us better define the scope of our paper and make our technical claims more rigorous. We hope the reviewer is satisfied with the latest revision we made to the presentation and thus can be supportive its publication.

REVIEWERS' COMMENTS

Reviewer #2 (Remarks to the Author):

Thank you for addressing my many comments. I appreciate your hard work and effort. I recommend the article to be published in its revised state.